# Agent-Centric Personalized Multiple Clustering with Multi-Modal LLMs

## Abstract

Personalized multiple clustering aims to generate diverse partitions of a dataset based on different user-specified aspects, rather than a single clustering. It has recently drawn research interest for accommodating varying user clustering preferences. Recent approaches primarily use CLIP embeddings with proxy learning to extract representations biased toward user interests. However, CLIP primarily focuses on coarse image-text alignment, lacking a deep contextual understanding of user interests. To overcome these limitations, we propose an agent-centric personalized clustering framework that leverages multi-modal large language models (MLLMs) as agents to comprehensively traverse a relational graph to search for clusters based on user interests. Due to the advanced reasoning mechanism of MLLMs, the obtained clusters align more closely with user-defined criteria than those obtained from CLIP-based representations. To reduce computational overhead, we shorten the agents' traversal path by constructing a relational graph using user-interest-biased embeddings extracted by MLLMs. A large number of weak edges can be filtered out based on embedding similarity, facilitating an efficient traversal search for agents. Experimental results show that the proposed method achieves NMI scores of 0.9667 and 0.9481 on the Card Order and Card Suits benchmarks, respectively, largely improving the SOTA model by over 140%. Code is available at https://anonymous.4open.science/r/Agent-Centric-Clustering-BEA5/.

## 1 Introduction

Clustering is a fundamental technique that partitions data into meaningful groups, uncovering underlying structures within a dataset. Traditional clustering methods Ng et al. (2001); Bishop & Nasrabadi (2006); Caron et al. (2018; 2020) rely on handcrafted features or fixed representations, which may fail to capture the inherent complex data relationships. Recent advances Lochman et al. (2024); Chu et al. (2024); Qian (2023); Ouldnoughi et al. (2023); Qian et al. (2022); Li et al. (2021); Duan et al. (2025) leverage learning-based techniques to obtain more expressive representations, leading to significant performance improvements. Most of these approaches produce only one data partition from a single perspective.

However, real-world data exhibit inherent complexity, making it impossible for any single clustering method to capture all relevant structures. Consequently, identifying multiple valid clusterings within a dataset is essential to address various analytical needs. For example, the fruits in Fig. 1 can be grouped by color or species, depending on the perspective. This need has driven the development of personalized multiple clustering, which aims to generate diverse partitions of a dataset based on different user clustering preferences, rather than a single fixed solution.

Typical multiple clustering methods Miklautz et al. (2020); Ren et al. (2022); Yao et al. (2023) primarily leverage autoencoders and data augmentation to extract diverse data representations, enabling clustering from different perspectives. However, identifying which clustering outcome aligns with user interests remains challenging, as these interests are often expressed through abstract keywords—such as color or species in the case of fruits, as illustrated in Fig. 1. Recent approaches Yao et al. (2024b;a) integrate CLIP Radford et al. (2021); Li et al. (2022; 2023; 2025) embeddings with proxy learning to obtain data representations tailored to user clustering preferences. While ef-

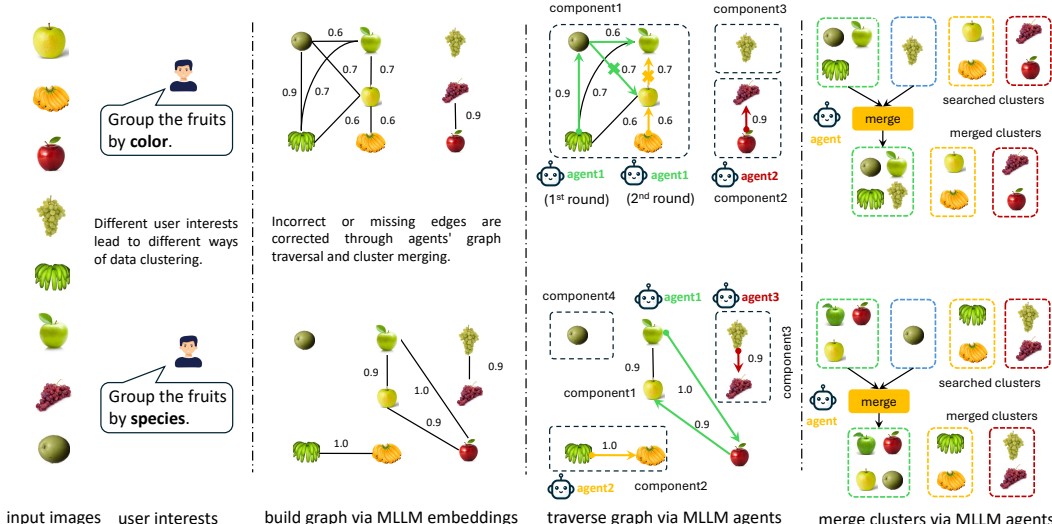

Figure 1: The workflow of the proposed agent-centric multiple clustering framework, which obtains a personalized clustering by using MLLMs as agents to traverse a relational graph based on user preferences. The relational graph is constructed from MLLM embeddings biased toward user interests.

fective, CLIP is trained on coarse-grained image-text pairs and is designed for overall cross-modal alignment, limiting its ability to focus solely on fine-grained, user-specified aspects.

To address these limitations, we propose an **agent-centric personalized multiple clustering framework** that employs shared multi-modal large language models (MLLMs) as agents to assess data relationships based on user interests and search for clusters accordingly. As shown in Fig. 1, we first construct a relational graph where nodes represent input images and edges encode pairwise similarities. Each agent is assigned to a connected component of the graph and initializes a cluster from the highest-degree node within that component. It then expands the cluster by traversing neighboring nodes, evaluating their membership based on user interests, and updating the cluster boundaries accordingly. Once a cluster is completed, the agent selects the next highest-degree unassigned node in the component to start a new cluster, repeating the traversal process until all nodes are assigned. After all agents complete, another agent performs a global review to merge semantically redundant clusters. Errors in graph connectivity are corrected through agent-based graph traversal and cluster merging. Leveraging the reasoning capabilities of MLLMs, our method yields clusters more faithfully aligned with user-defined criteria than those from CLIP-based approaches.

A naive approach to constructing the relational graph is to fully connect all nodes, resulting in an overly dense graph with many spurious edges. This significantly increases the traversal steps needed for agents to discard falsely connected neighbors, thereby compromising clustering efficiency. To address this, we build a weighted graph where edge weights reflect similarities between data embeddings extracted by MLLMs conditioned on user interests. By pruning weak edges based on similarity scores, we retain a compact set of high-quality neighbors, substantially reducing the traversal steps required for agents to evaluate them and improving clustering efficiency. Specifically, for each input image, the MLLM generates a textual description containing an embedding token based on user interests, where the hidden state of this token is projected as the data embedding. Unlike CLIP-based embeddings, which are limited by coarse-grained modality alignment, MLLM-based embeddings adaptively capture fine-grained, user-specified semantics through contextual reasoning, resulting in a sparser relational graph with primarily meaningful edges.

We evaluate our method on five publicly available visual datasets commonly used for multiple clustering tasks, and the results demonstrate its state-of-the-art performance. For instance, our method achieves NMI scores of $0.9667$ and $0.9481$ on the Card Order and Card Suits benchmarks, respectively, **surpassing the current state-of-the-art model by over** $140\%$. Our method provides a promising approach to unify personalized clustering and agent-based searching.

## 2 RELATED WORK

**Multiple Clustering** Multiple clustering explores diverse data partitions from different perspectives, gaining increasing attention. Early methods rely on hand-crafted rules and representations. For example, COALA Bae & Bailey (2006) generates new clusters using existing ones as a constraint, Hu et al. Hu et al. (2017) maximized eigengap across subspaces, and Dang et al. Dang & Bailey (2010) utilizes an expectation-maximization framework to optimize mutual information. Recent approaches leverage learning-based techniques for better representations. For instance, ENRC Miklautz et al. (2020) optimizes clustering objectives within a latent space learned by an auto-encoder, iMClusts Ren et al. (2022) leverages auto-encoders and multi-head attention to learn diverse feature representations, and AugDMC Yao et al. (2023) applies data augmentation to generate diverse image perspectives. However, it remains challenging to identify the clustering most relevant to user interests. Recently, Multi-MaP Yao et al. (2024b) and Multi-Sub Yao et al. (2024a) integrate CLIP embeddings with proxy learning to generate data representations aligned with user interests. While effective, CLIP lacks the deep contextual understanding necessary to capture abstract and nuanced user interests.

**LLM-Driven Agent Search** LLM-driven agent search integrates Large Language Models (LLMs) into search processes, enabling agents to reason, plan, and interact with tools for efficient information retrieval. For example, PaSa He et al. (2025) employs LLM-driven agents to search, read papers, and select references, ensuring comprehensive and accurate results for scholarly queries. Toolformer Schick et al. (2024) demonstrates how LLMs can interact with external APIs (e.g., search engines, calculators) to enhance factual accuracy. ReAct Yao et al. (2022) combines reasoning and acting, allowing agents to iteratively retrieve and process information for more effective decision-making. Recently, this field has shifted toward multi-agent search, emphasizing cooperative problem-solving, knowledge sharing, and decentralized decision-making. For example, CAMEL Li et al. introduces LLM agents with specialized roles (e.g., teacher and student) to refine search strategies collaboratively. Voyager Wang et al. (2023) applies LLMs in an open-world exploration setting (Minecraft), where agents autonomously collect, analyze, and apply new knowledge. In this paper, we apply MLLM-driven agents to search clusters based on user clustering preferences.

## 3 METHOD

In this section, we introduce our agent-centric personalized multiple clustering framework. We begin with an overview of the framework, followed by a detailed description of each component: agent-centric graph traversal, MLLM-based graph construction, and agent-based membership assessment.

### 3.1 OVERALL FRAMEWORK

The framework, illustrated in Fig. 2, begins by extracting image embeddings using an MLLM-based embedding extractor, tailoring representations to user interests. A relational graph is then constructed from these embeddings, where a large number of weak edges are filtered out based on embedding similarities. Next, multiple MLLM-based agents traverse this graph to search for clusters aligned with user preferences, with each agent specializing in a different connected component of the graph. After all agents complete, another agent performs a global review to merge semantically redundant clusters. This agent-centric approach produces clusters that better adhere to user-defined criteria than those obtained from CLIP-based representations.

### 3.2 AGENT-CENTRIC GRAPH TRAVERSAL

Here, we describe our agent-centric graph traversal approach for cluster discovery. We begin with a relational graph $\mathcal{G} = \{\mathcal{V}, \mathcal{E}, \mathcal{W}\}$ constructed as detailed in Sec 3.3, where each node $u \in \mathcal{V}$ represents an input image, and each edge $\{u, v\} \in \mathcal{E}$ carries a weight $w(u, v) \in \mathcal{W}$ indicating a precomputed, embedding-based similarity score between $u$ and $v$. Leveraging shared multi-modal large language models (MLLMs) as agents, we traverse $\mathcal{G}$ to search for clusters $\mathcal{S}$ aligned with user interests $T$.

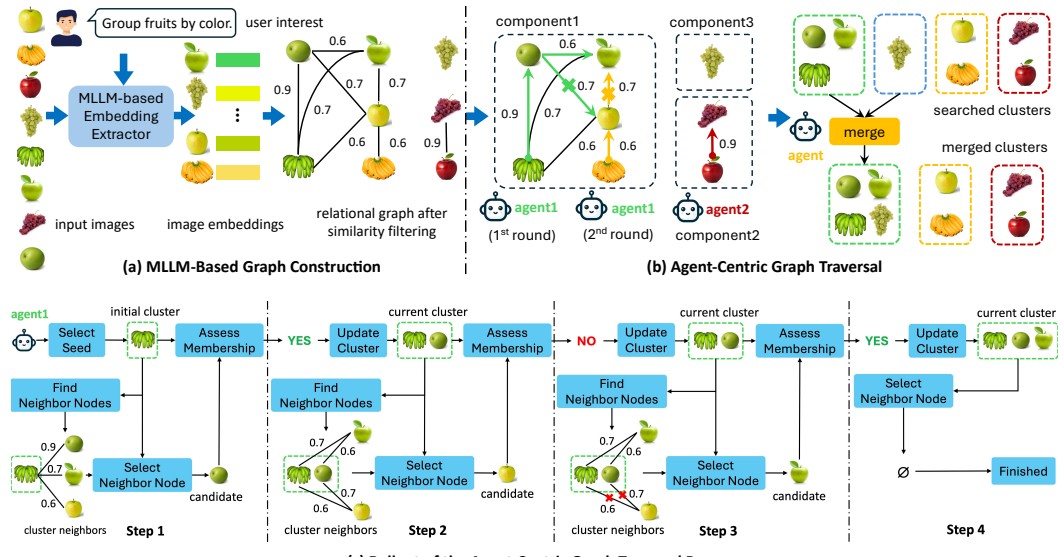

Figure 2: Overview of the Agent-Centric Personalized Multiple Clustering Framework. (a) MLLM-based graph construction, where image embeddings are extracted using MLLM based on user interests, from which a relational graph is constructed. (b) Agent-centric graph traversal, where agents search for clusters by traversing the graph. (c) Rollout of the graph traversal process, where agents expand cluster iteratively by assessing neighboring nodes based on user-defined criteria.

As outlined in Algorithm 1, we first identify the connected components of the relational graph $\mathcal{G}$, resulting in a total of $M$ components:

$$\{\mathcal{C}_i\}_{i=1}^M = \text{ConnectedComponents}(\mathcal{G}). \tag{1}$$

For each connected component $\mathcal{C}_i$, we select the highest-degree node $v_i^*$ and designate it as the seed for an initial cluster $\mathcal{S}_i$ as follows:

$$v_i^* = \arg\max_{v \in \mathcal{C}_i} \sum_{u \in \mathcal{N}(v)} w(v, u), \quad \mathcal{S}_i = \{v_i^*\}, \tag{2}$$

where $\mathcal{N}(v)$ is the neighborhood of node $v$. We then assign an MLLM-based agent $\mathcal{A}_i$ to each initial cluster $\mathcal{S}_i$, resulting in a total of $M$ agents, denoted as $\{\mathcal{A}_i\}_{i=1}^M$. Each agent $\mathcal{A}_i$ iteratively expands its corresponding cluster $\mathcal{S}_i$ by evaluating the membership of its neighboring nodes in $\mathcal{S}_i$ based on user interests $T$. As shown in Fig. 2 (c) and Algorithm 1, at each step, the agent $\mathcal{A}_i$ first determines the neighborhood of the current cluster $\mathcal{S}_i$, defined as:

$$\mathcal{N}(\mathcal{S}_i) = \bigcup_{v \in \mathcal{S}_i} \mathcal{N}(v) \setminus \mathcal{S}_i. \tag{3}$$

Next, the agent selects a candidate neighboring node $v_j^*$ from $\mathcal{N}(\mathcal{S}_i)$ that exhibits the highest weighted connectivity to $\mathcal{S}_i$, given by:

$$v_j^* = \arg\max_{v \in \mathcal{N}(\mathcal{S}_i)} \sum_{u \in \mathcal{S}_i} w(v, u). \tag{4}$$

The agent then assesses whether $v_j^*$ should be included in $\mathcal{S}_i$ based on user-defined criteria $T$, as detailed in Sec. 3.4. If the candidate node $v_j^*$ is deemed a valid member, it is merged into the current cluster $\mathcal{S}_i$, updating $\mathcal{S}_i$ as follows:

$$\mathcal{S}_i \leftarrow \mathcal{S}_i \cup \{v_j^*\}. \tag{5}$$

Otherwise, the edges between the candidate node $v_j^*$ and the current cluster $\mathcal{S}_i$ are removed from the edge set $\mathcal{E}$, yielding an updated edge set:

$$\mathcal{E} \leftarrow \mathcal{E} \setminus \{(u, v_j^*) \in \mathcal{E} \mid u \in \mathcal{S}_i\}. \tag{6}$$

---

**Algorithm 1:** Agent-Centric Graph Traversal for Personalized Multiple Clustering

---

**Input:** Relational graph $\mathcal{G} = (\mathcal{V}, \mathcal{E}, \mathcal{W})$, user interests $T$
**Output:** Final set of clusters $\mathcal{S}$

1 Initialize $\mathcal{S} \leftarrow \emptyset$
2 $\{\mathcal{C}_1, \mathcal{C}_2, \ldots, \mathcal{C}_M\} \leftarrow \texttt{ConnectedComponents}(\mathcal{G})$
3 **for** $i \leftarrow 1$ **to** $M$ **do**
4     **while** $\mathcal{C}_i \neq \emptyset$ **do**
5         $v_i^* \leftarrow \operatorname{argmax}_{v \in \mathcal{C}_i} \sum_{u \in \mathcal{N}(v)} w(v, u); \quad \mathcal{S}_i \leftarrow \{v_i^*\}$         // Initialize cluster
6         $\mathcal{N}(\mathcal{S}_i) \leftarrow \bigcup_{v \in \mathcal{S}_i} \mathcal{N}(v) \setminus \mathcal{S}_i$
7         **while** $\mathcal{N}(\mathcal{S}_i) \neq \emptyset$ **do**
8             $v_j^* \leftarrow \operatorname{argmax}_{v \in \mathcal{N}(\mathcal{S}_i)} \sum_{u \in \mathcal{S}_i} w(v, u)$
9             **if** *agent $\mathcal{A}_i$ determines $v_j^*$ belongs to $\mathcal{S}_i$ based on $T$* **then**
10                 $\mathcal{S}_i \leftarrow \mathcal{S}_i \cup \{v_j^*\}$
11             **else**
12                 $\mathcal{E} \leftarrow \mathcal{E} \setminus \{(u, v_j^*) \in \mathcal{E} \mid u \in \mathcal{S}_i\}$
13             $\mathcal{N}(\mathcal{S}_i) \leftarrow \bigcup_{v \in \mathcal{S}_i} \mathcal{N}(v) \setminus \mathcal{S}_i$
14         $\mathcal{C}_i \leftarrow \mathcal{C}_i \setminus \mathcal{S}_i; \quad \mathcal{S} \leftarrow \mathcal{S} \cup \{\mathcal{S}_i\}$
15 **while** $\mathcal{S}$ *changed* **do**
16     **foreach** $(\mathcal{S}_p, \mathcal{S}_q) \in \binom{\mathcal{S}}{2}$ **do**
17         **if** *agent $\mathcal{A}_{merge}$ decides to merge $\mathcal{S}_p$ and $\mathcal{S}_q$ based on $T$* **then**
18             $\mathcal{S}_{pq} \leftarrow \mathcal{S}_p \cup \mathcal{S}_q; \quad \mathcal{S} \leftarrow \left(\mathcal{S} \setminus \{\mathcal{S}_p, \mathcal{S}_q\}\right) \cup \{\mathcal{S}_{pq}\}; \quad$ **break**
19 **return** $\mathcal{S}$

---

The process from Eq. (3) to Eq. (6) is repeated until the neighborhood $\mathcal{N}(\mathcal{S}_i)$ is empty, determining the cluster $\mathcal{S}_i$. The agent then removes the nodes in $\mathcal{S}_i$ from the component $\mathcal{C}_i$ as follows:

$$\mathcal{C}_i \leftarrow \mathcal{C}_i \setminus \mathcal{S}_i. \tag{7}$$

Next, the agent selects the highest-degree unassigned node in $\mathcal{C}_i$ to initiate a new cluster, repeating the traversal process from Eq. (2) to Eq. (7) until all nodes in $\mathcal{C}_i$ are assigned. The $M$ agents, $\{\mathcal{A}_i\}_{i=1}^M$, traverse the graph $\mathcal{G}$ in parallel, with the searched clusters collected into the set $\mathcal{S}$. Finally, another agent performs a global review to merge semantically redundant clusters in $\mathcal{S}$ based on user interests $T$. Incorrect or missing edges in the graph $\mathcal{G}$ are corrected through agent-based graph traversal and cluster merging. Leveraging the contextual reasoning capabilities of MLLMs, our method yields clusters more faithfully aligned with user interests compared to CLIP-based approaches.

### 3.3 MLLM-BASED GRAPH CONSTRUCTION

Here, we detail the construction of the relational graph $\mathcal{G} = \{\mathcal{V}, \mathcal{E}, \mathcal{W}\}$ used for agent traversal. Each node $u \in \mathcal{V}$ corresponds to an input image. Each edge $(u, v) \in \mathcal{E}$ is assigned a weight $w(u, v) \in W$, determined by the similarity between the embeddings $\mathcal{H}(u)$ and $\mathcal{H}(v)$. These embeddings are extracted by the MLLM for nodes $u$ and $v$ based on user interests $T$.

As shown in Fig. 3 (a), we first design an embedding generation instruction based on user interests $T$. For example, if the user intends to cluster fruit images by "color", the instruction is formulated as: "Describe the color of the fruit in the provided image in detail and generate <embedding> based on the description." We then provide the image $u$ and the instruction as inputs to the MLLM, which responds with a reasoning statement followed by an <embedding> token. The hidden state of the <embedding> token is then projected to obtain the user-interest-biased embedding $\mathcal{H}(u)$. After obtaining the embeddings $\mathcal{H}(u)$ and $\mathcal{H}(v)$, the edge weight $w(u, v)$ is computed as:

$$w(u, v) = \sigma(\beta \cdot \mathcal{H}(u)^\top \mathcal{H}(v)), \tag{8}$$

where $\beta$ is a learnable logit scale, and $\sigma$ denotes the sigmoid function. An edge $(u, v)$ is retained in the edge set $\mathcal{E}$ only if its weight $w(u, v)$ exceeds a predefined threshold $\tau$.

As shown in Fig. 3 (b), for training data generation, we perform hard negative mining to select the top $1,024$ image pairs with the highest similarity entropy in each epoch. GPT-4 Achiam et al. (2023)

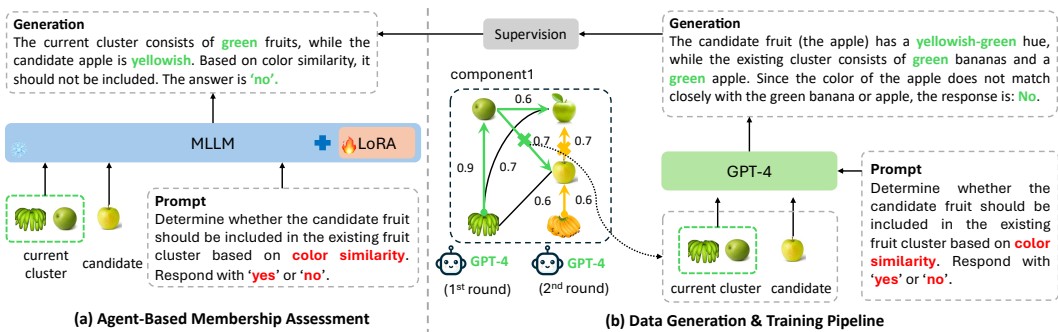

Figure 3: Illustration of image embedding extraction using MLLM based on user interests.

Figure 4: Illustration of Agent-based assessment of candidate membership according to user interests.

evaluates the similarity of these pairs based on user interests $T$, from which binary pseudo labels are inferred. It also generates detailed descriptions for each image, with an appended `<embedding>` token. During training, the MLLM produces descriptions and embeddings for each image pair conditioned on $T$. We compute embedding similarities following Eq. (8) and supervise them using GPT-4's pseudo labels via binary cross-entropy loss. The generated descriptions are aligned with GPT-4's outputs using cross-entropy loss. The MLLM is finetuned with LoRA Hu et al. (2022).

## 3.4 AGENT-BASED MEMBERSHIP ASSESSMENT

Here, we describe how the agent $\mathcal{A}_i$ assesses whether a candidate node $v_j^*$ should be included in the current cluster $\mathcal{S}_i$ based on user interests $T$. As illustrated in Fig. 4 (a), we first select the top-$K$ nodes with the highest degrees within $\mathcal{S}_i$ to form a set of representative nodes:

$$\mathcal{R}_i = \arg \max_{\substack{R \subseteq \mathcal{S}_i \\ |\mathcal{R}| = min(K, |\mathcal{S}|)}} \sum_{v \in R} \sum_{u \in \mathcal{N}(v) \cap \mathcal{S}_i} w(v, u), \qquad (9)$$

where $\mathcal{R}_i$ serves as a compact representation of the current cluster $\mathcal{S}_i$. We then design a membership assessment instruction tailored to user interests $T$. For instance, if the user aims to cluster fruit images by "color", the instruction is: "Determine whether the candidate fruit should be included in the existing fruit cluster based on color similarity. Respond with 'yes' or 'no'." The candidate node $v_j^*$ and the representative node set $\mathcal{R}_i$ are provided to the agent $\mathcal{A}_i$ along with this instruction. The agent responds with a reasoning statement followed by a binary decision. Based on the agent's assessment, we update the current cluster $\mathcal{S}_i$ as described in Eqs. (5) and (6).

As shown in Fig. 4 (b), for training data generation, we begin by constructing a relational graph using embeddings extracted by the MLLM. We then uniformly sample a subgraph containing $1,024$ nodes from the relational graph. GPT-4 traverses this subgraph according to Eqs. (1)–(7), assessing the membership of each neighboring node with respect to the current cluster based on user interests $T$,

Table 1: Comparison with state-of-the-art methods across multiple clustering benchmarks.

| | | Fruit | | Fruit360 | | Card | | CMUface | | | | CIFAR10-MC | |
| | | Color | Species | Color | Species | Order | Suits | Emotion | Sunglass | Identity | Pose | Type | Environment |
|---|---|---|---|---|---|---|---|---|---|---|---|---|---|
| MSC | NMI | 0.6886 | 0.1627 | 0.2544 | 0.2184 | 0.0807 | 0.0497 | 0.1284 | 0.1420 | 0.3892 | 0.3687 | 0.1547 | 0.1136 |
| | RI | 0.8051 | 0.6045 | 0.6054 | 0.5805 | 0.7805 | 0.3587 | 0.6736 | 0.5745 | 0.7326 | 0.6322 | 0.3296 | 0.3082 |
| MCV | NMI | 0.6266 | 0.2733 | 0.3776 | 0.2985 | 0.0792 | 0.0430 | 0.1433 | 0.1201 | 0.4637 | 0.3254 | 0.1618 | 0.1379 |
| | RI | 0.7685 | 0.6597 | 0.6791 | 0.6176 | 0.7128 | 0.3638 | 0.5268 | 0.4905 | 0.6247 | 0.6028 | 0.3305 | 0.3344 |
| ENRC | NMI | 0.7103 | 0.3187 | 0.4264 | 0.4142 | 0.1225 | 0.0676 | 0.1592 | 0.1493 | 0.5607 | 0.2290 | 0.1826 | 0.1892 |
| | RI | 0.8511 | 0.6536 | 0.6868 | 0.6984 | 0.7313 | 0.3801 | 0.6630 | 0.6209 | 0.7635 | 0.5029 | 0.3469 | 0.3599 |
| iMClusts | NMI | 0.7351 | 0.3029 | 0.4097 | 0.3861 | 0.1144 | 0.0716 | 0.0422 | 0.1929 | 0.5109 | 0.4437 | 0.2040 | 0.1920 |
| | RI | 0.8632 | 0.6743 | 0.6841 | 0.6732 | 0.7658 | 0.3715 | 0.5932 | 0.5627 | 0.8260 | 0.6114 | 0.3695 | 0.3664 |
| AugDMC | NMI | 0.8517 | 0.3546 | 0.4594 | 0.5139 | 0.1440 | 0.0873 | 0.0161 | 0.1039 | 0.5875 | 0.1320 | 0.2855 | 0.2927 |
| | RI | 0.9108 | 0.7399 | 0.7392 | 0.7430 | 0.8267 | 0.4228 | 0.5367 | 0.5361 | 0.8334 | 0.5517 | 0.4516 | 0.4689 |
| DDMC | NMI | 0.8973 | 0.3764 | 0.4981 | 0.5292 | 0.1563 | 0.0933 | 0.1726 | 0.2261 | 0.6360 | 0.4526 | 0.3991 | 0.3782 |
| | RI | 0.9383 | 0.7621 | 0.7472 | 0.7703 | 0.8326 | 0.6469 | 0.7593 | 0.7663 | 0.8907 | 0.7904 | 0.5827 | 0.5547 |
| Multi-Map | NMI | 0.8619 | 1.0000 | 0.6239 | 0.5284 | 0.3653 | 0.2734 | 0.1786 | 0.3402 | 0.6625 | 0.4693 | 0.4969 | 0.4598 |
| | RI | 0.9526 | 1.0000 | 0.8243 | 0.7582 | 0.8587 | 0.7039 | 0.7105 | 0.7068 | 0.9496 | 0.6624 | 0.7104 | 0.6737 |
| Multi-Sub | NMI | 0.9693 | 1.0000 | 0.6654 | 0.6123 | 0.3921 | 0.3104 | 0.2053 | 0.4870 | 0.7441 | 0.5923 | 0.5271 | 0.4828 |
| | RI | 0.9964 | 1.0000 | **0.8821** | **0.8504** | 0.8842 | 0.7941 | **0.8527** | 0.8324 | **0.9834** | 0.8736 | 0.7394 | 0.7096 |
| **Ours** | NMI | **1.0000** | **1.0000** | **0.7214** | **0.6532** | **0.9667** | **0.9481** | 0.2196 | **0.9720** | 0.7631 | **0.7789** | **0.6385** | **0.5931** |
| | RI | **1.0000** | **1.0000** | 0.8715 | 0.8436 | **0.9952** | **0.9882** | 0.8415 | **0.9936** | 0.9763 | **0.9002** | **0.7812** | **0.7439** |

as defined in Eq. 9. We collect these assessments at each traversal step as training samples. During training, the MLLM predicts membership for each sampled neighboring node and its corresponding cluster based on $T$. These predictions are supervised using GPT-4's assessments via cross-entropy loss. The MLLM is finetuned using LoRA Hu et al. (2022).

# 4 EXPERIMENT

## 4.1 DATASETS

We evaluate our method on all publicly available multiple-clustering benchmarks: Card Yao et al. (2023), CMUface Günnemann et al. (2014), Fruit Hu et al. (2017), Fruit360 Yao et al. (2023), and CIFAR10-MC Yao et al. (2024a). The Card dataset contains 8,029 playing card images with two clustering criteria: order (Ace–King) and suits (clubs, diamonds, hearts, spades). CMUface includes 640 facial images annotated for pose (left, right, straight, up), identity (20 individuals), sunglasses (with/without), and emotion (angry, sad, happy, neutral). Fruit consists of 105 images clusterable by species (apple, banana, grape) or color (green, red, yellow). Fruit360 extends this with 4,856 images labeled by species (apple, banana, grape, cherry) and color (green, red, yellow, burgundy). CIFAR10-MC includes 60,000 images grouped by type (transportation, animals) and environment (land, air, water).

## 4.2 IMPLEMENTATION DETAILS

We adopt LLaVA Liu et al. (2024) as our MLLM model, using Qwen2-7B Yang et al. (2024) as the language model. We fine-tune only the language model with LoRA Hu et al. (2022). Both the embedding extractor and the agent are trained for 50 epochs on 4 A100 GPUs over 3 days, with a total batch size of 32. Optimization is performed using AdamW Loshchilov & Hutter (2017) with a learning rate of $1e-5$ and a cosine decay schedule. We set the embedding similarity threshold $\tau$ to 0.6 to filter weak edges. Clustering performance is evaluated using Normalized Mutual Information (NMI) White et al. (2004) and Rand Index (RI) Rand (1971).

## 4.3 RESULTS

As shown in Table 1, our method consistently outperforms state-of-the-art models across all multiple clustering benchmarks. On the Fruit dataset, it achieves perfect NMI and RI scores (1.0000) for both color- and species-based clustering. On Fruit360, it attains an NMI of 0.7214 for color-based clustering, exceeding Multi-Map and Multi-Sub by 15.6% and 8.4%, respectively, and an NMI of 0.6532 for species-based clustering, outperforming them by 24.3% and 6.7%. On CIFAR10-MC, our method achieves an NMI of 0.6385 and RI of 0.7812 for type-based clustering, surpassing Multi-Sub by 21.1% and 5.7%, and an NMI of 0.5931 and RI of 0.7439 for environment-based clustering,

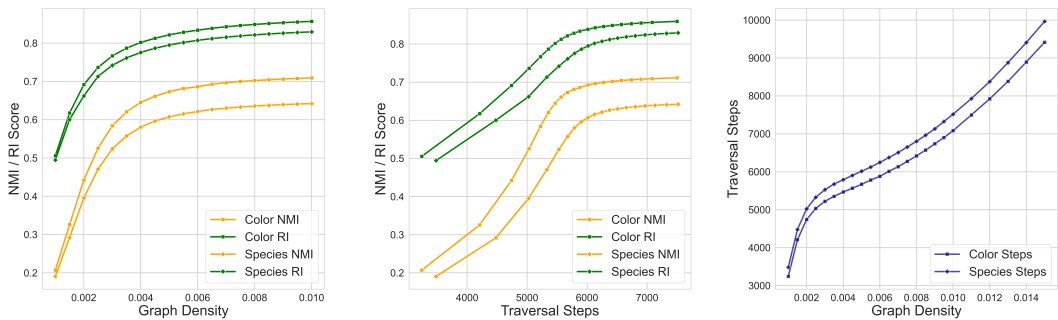

Figure 5: Graph density vs. clustering metrics and traversal steps.

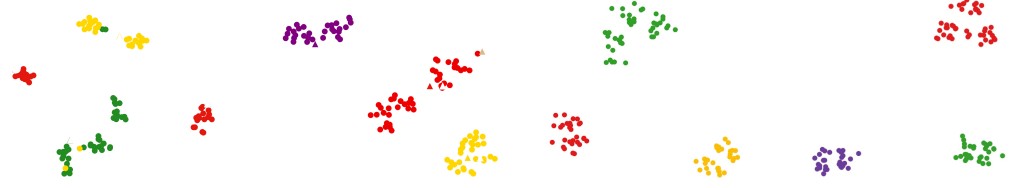

(1) Color of CLIP-based Emb    (2) Species of CLIP-Based Emb    (3) Color of MLLM-Based Emb    (4) Species of MLLM-Based Emb

Figure 6: T-SNE visualization of MLLM-based and CLIP-based embeddings on the Fruit dataset.

exceeding Multi-Sub by $22.8\%$ and $4.8\%$. On Card, it reaches $0.9667$ NMI and $0.9952$ RI for order-based clustering, and $0.9481$ NMI and $0.9882$ RI for suits-based clustering, outperforming Multi-Sub by up to $205.4\%$ in NMI and $24.4\%$ in RI. On CMUface, it achieves $0.9720$ NMI and $0.9936$ RI for sunglass-based clustering, and $0.7789$ NMI and $0.9002$ RI for pose-based clustering, showing up to $99.6\%$ and $31.5\%$ improvements over Multi-Sub.

These significant improvements highlight the effectiveness of our agent-centric clustering framework in capturing user interests (e.g., order, suits, sunglass, pose) with advanced contextual understanding capabilities from MLLMs, while minimizing interference from irrelevant information.

## 4.4 ABLATION STUDIES

**Density of Relational Graph.** We investigate how graph density affects agent traversal by varying the embedding similarity threshold $\tau$ to construct relational graphs of different densities. Experiments are conducted on Fruit360. As shown in Fig. 5, increasing graph density (i.e., relaxing $\tau$) improves NMI and RI for both color- and species-based clustering until performance saturates. For instance, Color RI and Species RI quickly rise and plateau around $0.87$ and $0.84$, while Color NMI and Species NMI converge near $0.72$ and $0.65$. This suggests that once sufficient edges form connected components of semantically similar nodes, further densification yields diminishing returns.

However, denser graphs incur higher traversal costs. As graph density increases from $0.001$ to $0.015$, the number of agent traversal steps (dark blue curve) rises from $3,000$ to nearly $10,000$ due to a significant increase in spurious edges, indicating substantial computational overhead. A moderate $\tau$ strikes a balance: clustering accuracy stabilizes while reducing unnecessary traversal. These findings underscore the benefits of using MLLM-based embeddings to construct relational graphs. By pruning weak edges with low similarity scores, agents avoid exhaustive searches in irrelevant regions. At the same time, the context-aware nature of MLLM embeddings retains high-quality edges, aligning clusters with user interests while keeping traversal efficient.

**MLLM-based vs. CLIP-based Embeddings.** We compare our MLLM-based embeddings with CLIP-based ones across various clustering methods. We use Multi-Map Yao et al. (2024b) as the CLIP baseline, as both approaches decouple embedding extraction from clustering for fair ablation. As shown in Table 2, MLLM-based embeddings consistently outperform CLIP-based ones. For example, on CMUface ("Sunglass" aspect), MLLM-based embeddings improve NMI from $0.3402$, $0.3952$, and $0.8456$ to $0.7193$, $0.7698$, and $0.9720$ for K-means, HDBSCAN, and agent-centric clustering, respectively—demonstrating superior alignment with user-defined criteria. Fig. 6 further

Table 2: Ablation study on the Fruit360, Card, and CMUface datasets.

| Embedding | Dataset | Aspect | K-Means | | HDBSCAN | | Agent-Centric Graph Traversal | | |
|---|---|---|---|---|---|---|---|---|---|
| | | | NMI | RI | NMI | RI | NMI | RI | Traversal Steps |
| **CLIP Emb** | Fruit360 | Color | 0.6239 | 0.8243 | 0.6541 | 0.8408 | 0.6922 | 0.8573 | 14821 |
| | | Species | 0.5284 | 0.7582 | 0.5702 | 0.7851 | 0.6158 | 0.8180 | 14712 |
| | Card | Order | 0.3653 | 0.8587 | 0.4163 | 0.8708 | 0.8464 | 0.9679 | 29456 |
| | | Suits | 0.2734 | 0.7039 | 0.3368 | 0.7275 | 0.8132 | 0.9313 | 14969 |
| | CMUface | Sunglass | 0.3402 | 0.7068 | 0.3952 | 0.7301 | 0.8456 | 0.9362 | 1281 |
| | | Pose | 0.4693 | 0.6624 | 0.4935 | 0.6804 | 0.7168 | 0.8526 | 4015 |
| **MLLM Emb** | Fruit360 | Color | 0.6629 | 0.8432 | 0.6912 | 0.8578 | **0.7214** | **0.8715** | **6404** |
| | | Species | 0.5783 | 0.7924 | 0.6138 | 0.8179 | **0.6532** | **0.8436** | **6964** |
| | Card | Order | 0.7262 | 0.9406 | 0.7743 | 0.9515 | **0.9667** | **0.9952** | **12837** |
| | | Suits | 0.6782 | 0.8745 | 0.7322 | 0.8972 | **0.9481** | **0.9882** | **9970** |
| | CMUface | Sunglass | 0.7193 | 0.9005 | 0.7698 | 0.9191 | **0.9720** | **0.9936** | **798** |
| | | Pose | 0.6551 | 0.8051 | 0.6799 | 0.8241 | **0.7789** | **0.9002** | **1452** |

Table 3: Ablation study on number of representative nodes $K$.

| Num of Representatives | Fruit360-Color | | Fruit360-Species | | Card-Order | | Card-Suits | |
|---|---|---|---|---|---|---|---|---|
| | NMI | RI | NMI | RI | NMI | RI | NMI | RI |
| $K = 1$ | 0.7131 | 0.8643 | 0.6424 | 0.8361 | 0.9625 | 0.9930 | 0.9421 | 0.9860 |
| $K = 2$ | 0.7185 | 0.8691 | 0.6490 | 0.8402 | 0.9652 | 0.9943 | 0.9461 | 0.9873 |
| $K = 3$ | 0.7214 | 0.8715 | 0.6532 | 0.8436 | 0.9667 | 0.9952 | 0.9481 | 0.9882 |
| $K = 4$ | 0.7240 | 0.8730 | 0.6559 | 0.8457 | 0.9673 | 0.9957 | 0.9492 | 0.9888 |

illustrates this: MLLM-based embeddings yield compact, well-separated clusters in t-SNE visualization on the Fruit dataset, while CLIP-based embeddings show higher intra-cluster variance, particularly in color-based clustering. These results confirm that MLLM embeddings more effectively capture user interests (e.g., color, species), leading to improved clustering accuracy.

Moreover, MLLM-based embeddings significantly reduce agent traversal steps in agent-centric clustering. On Fruit360 ("Color" aspect), agent traversal completes in $6,404$ steps with MLLM embeddings, compared to $14,821$ with CLIP-based ones. This efficiency arises from the ability of MLLM embeddings to better prune noisy edges, retaining a compact set of high-quality neighbors and reducing the number of membership assessments needed during graph traversal.

**Agent-Centric Clustering vs. K-Means & HDBSCAN.** We compare our agent-centric graph traversal clustering with classical K-Means and HDBSCAN. As shown in Table 2, the agent-centric approach consistently outperforms both baselines in NMI and RI across all datasets and embedding types. On Card ("Order" aspect) with MLLM-based embeddings, it achieves an NMI of $0.9667$, outperforming K-Means ($0.7262$) and HDBSCAN ($0.7743$). The improvement is even larger with CLIP-based embeddings, reaching an NMI of $0.8464$ vs. $0.3653$ (K-Means) and $0.4163$ (HDBSCAN), demonstrating the robustness of the agent-centric approach. Furthermore, it requires fewer traversal steps with MLLM-based embeddings than with CLIP-based ones, improving efficiency.

**Number of Representative Nodes.** Table 3 shows the effect of varying $K$ on clustering performance (NMI and RI). Using a single representative node may fail to capture the cluster's diversity, leading to lower performance. Increasing $K$ improves both metrics by better reflecting the cluster's structure, although gains beyond three or four representatives are typically modest. Thus, a moderate $K$ strikes the best balance between capturing diversity and avoiding redundant information.

## 5 CONCLUSION

This paper presents an agent-centric clustering framework that leverages MLLMs to traverse relational graphs and search for clusters based on user interests. By leveraging MLLMs' reasoning abilities, our method better captures user clustering preferences than CLIP-based approaches. To improve efficiency, we construct relational graphs from user-interest-biased embeddings derived from MLLMs, and reduce agents' traversal steps by filtering weak connections. Experiments on multiple benchmarks demonstrate the state-of-the-art performance of our method, validating its effectiveness.

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
