APPENDIX

## A   ACKNOWLEDGMENT OF LLM USAGE

During the preparation of this manuscript, large language models (LLMs) were used exclusively for language editing purposes, including correcting typographical errors, improving grammar, and refining phrasing. LLMs were not used for generating research ideas, performing analyses, producing results, or interpreting findings. The authors take full responsibility for all scientific content presented in this work.

The Appendix is organized as follows:

- Appendix B: Additional implementation details of our proposed framework, including key code snippets and explanations for each core component:
  - **Embedding & Agent Training Data Generation** (B.1): Describes how to generate labeled data for embedding and agent training with GPT-4.
  - **Multi-Modal Large Language Model** (B.2): Describes the initialization and LoRA-based fine-tuning of the MLLM used for both embedding extraction and agent-based decision making.
  - **MLLM-Based Image Embedding Extraction** (B.3): Details the process of generating user-interest-biased image embeddings by extracting hidden states associated with the `<embedding>` token.
  - **MLLM-Based Image Embedding Training** (B.4): Explains how image embeddings are fine-tuned using GPT-4-generated captions and similarity labels, supervised with cross-entropy and contrastive loss.
  - **MLLM-Embedding-Based Graph Construction** (B.5): Outlines the construction of a sparse relational graph using MLLM embeddings and a similarity threshold, to guide agent traversal.
  - **Agent-Based Node Membership & Cluster Merge Assessment** (B.6): Presents the use of MLLM-generated textual conclusions to assess candidate node membership and determine whether clusters should be merged.
  - **Agent-Centric Graph Traversal** (B.7): Describes how agents iteratively expand clusters within each connected component based on MLLM-guided evaluations of neighboring nodes.
  - **Agent-Centric Cluster Merge** (B.8): Details the global refinement stage in which semantically redundant clusters are merged through agent-based inter-cluster similarity assessments.
- Appendix C: Additional qualitative examples that highlight the interpretability and effectiveness of our agent-based clustering framework.
  - **Examples of Agent-Based Node Membership Assessment** (C.1): Showcases how agents assess candidate nodes across different clustering aspects (e.g., number, suits, color) by reasoning over visual features and user-defined criteria. Incorrect connections in the relational graph are progressively pruned through these assessments, improving clustering purity.
  - **Examples of Agent-Based Cluster Merge Assessment** (C.2): Demonstrates how agents evaluate whether semantically similar clusters should be merged. The examples illustrate how missing edges in the graph are recovered via agent-guided assessments, enhancing the cohesion and completeness of the final clustering results.

## B   MORE FRAMEWORK DETAILS WITH KEY CODE SNIPPETS

### B.1   EMBEDDING & AGENT TRAINING DATA GENERATION

The embedding and agent training data are generated dynamically during the training process. For embedding training, in each epoch, we first extract user-interest-biased image embeddings using the MLLM, as described in Sec.B.3. We then perform hard negative mining by selecting the top

$1,024$ image pairs with the highest embedding similarity entropy. Over 50 training epochs, this results in approximately $50,000$ image pairs. As shown in the provided code snippet, GPT-4 is used to evaluate the semantic similarity of these image pairs based on user-defined criteria, from which binary pseudo labels are inferred. These labels supervise the pairwise embedding similarities using a binary cross-entropy loss, as detailed in Sec.B.4. In addition, GPT-4 generates detailed captions for each image, appending a special <embedding> token at the end. These captions serve as targets for training the MLLM's language generation via cross-entropy loss. The hidden state corresponding to the <embedding> token in the MLLM output is then projected to form the final image embedding, which captures user-interest-aligned semantics.

For agent training, we construct a relational graph in each epoch using embeddings extracted by the MLLM, as described in Sec.B.5. From this graph, we uniformly sample a subgraph containing $1,024$ nodes. GPT-4 then traverses this subgraph following the procedure in Sec.B.7, evaluating the membership of each neighboring node with respect to the current cluster based on user-defined interests. These membership assessments are collected at each traversal step and used as training samples. The membership assessment with GPT4 is similar to "generate_image_similarity_with_GPT4" in the following code snippet. During training, the MLLM is tasked with predicting whether each sampled neighboring node belongs to its associated cluster, conditioned on user interests, as described in Sec. B.6. The model's predictions are supervised using GPT-4's decisions, optimized via cross-entropy loss.

```python
openai_api_key = "xxx"

prompt_caption_color = "Describe the color of the fruit in the provided
    image in detail."

prompt_similarity_color = "Determine whether the two fruits share the
    same color. Respond with <CONCLUSION> YES </CONCLUSION> or <
    CONCLUSION> NO </CONCLUSION>."

def encode_image_base64(path):
    with open(path, "rb") as image_file:
        return base64.b64encode(image_file.read()).decode("utf-8")

def generate_image_caption_with_GPT4(image_path):
    base64_image = encode_image_base64(image_path)
    client = openai.OpenAI(api_key=openai_api_key)

    response = client.chat.completions.create(
        model="gpt-4o",
        messages=[
            {
                "role": "user",
                "content": [
                    {"type": "text", "text": prompt_caption_color},
                    {
                        "type": "image_url",
                        "image_url": {
                            "url": f"data:image/png;base64,{base64_image}
    "
                        }
                    }
                ]
            }
        ]
    )
    caption = response.choices[0].message.content + "Embedding: <
    embedding>"
    return caption

def generate_image_similarity_with_GPT4(image_path1, image_path2)
    base64_image1 = encode_image_base64(image_path1)
    base64_image2 = encode_image_base64(image_path2)
    client = openai.OpenAI(api_key=openai_api_key)
```

```
      response = client.chat.completions.create(
          model="gpt-4o",
          messages=[
              {
                  "role": "user",
                  "content": [
                      {"type": "text", "text": prompt_similarity_color},
                      {
                          "type": "image_url",
                          "image_url": {
                              "url": f"data:image/png;base64,{base64_image1
      }",
                              "detail": "auto"
                          }
                      },
                      {
                          "type": "image_url",
                          "image_url": {
                              "url": f"data:image/jpeg;base64,{
      base64_image2}",
                              "detail": "auto"
                          }
                      }
                  ]
              }
          ],
          max_tokens=500
      )
      img_similarity_reason = response.choices[0].message.content
      match = re.search(r"<CONCLUSION>(.*?)</CONCLUSION>",
      img_similarity_reason, re.DOTALL)
      if match:
          conclusion = match.group(1).strip()
          if "yes" in conclusion.lower():
              img_similarity_label = 1.0
          elif "no" in conclusion.lower():
              img_similarity_label = 0.0
          else:
              img_similarity_label = -1.0
              print(f"No answer found in {conclusion}.")
      else:
          img_similarity_label = -1.0
          print(f"No conclusion found in {output}.")
      return img_similarity_label

caption_dict = {}
similarity_dict = {}

def generate_embedding_training_data_with_GPT4(sample_pairs):
    targets = []
    for image_path1, image_path2 in sample_pairs:
        if image_path1 not in caption_dict:
            caption_dict[image_path1] = generate_image_caption_with_GPT4(
    image_path1)
        if image_path2 not in caption_dict:
            caption_dict[image_path2] = generate_image_caption_with_GPT4(
    image_path2)
        if (image_path1, image_path2) not in similarity_dict:
            similarity_dict[(image_path1, image_path2)] =
    generate_image_similarity_with_GPT4(image_path1, image_path2)

        target = {
            "caption1": caption_dict[image_path1],
```

```
            "caption2": caption_dict[image_path2],
            "img_similarity_label": similarity_dict[(image_path1,
    image_path2)],
        }
        targets.append(target)
    return targets
```

Listing 1: Embedding Training Data Generation Code Snippet

## B.2 MULTI-MODAL LARGE LANGUAGE MODEL

The Multi-Modal Large Language Model (MLLM) serves as a crucial component in the proposed agent-centric personalized multiple clustering framework. In this setup, the MLLM is used both as an embedding extractor and an agent, assisting in the traversal of relational graphs to search for clusters based on user-specific interests. As detailed in the provided code snippet, the model is initialized with a pre-trained version of LlavaQwenForCausalLM, incorporating a vision tower for multimodal capabilities, enabling it to generate embeddings that reflect both visual and textual data.

A distinctive feature of the approach is the fine-tuning of the language model using LoRA, which ensures that only specific layers, including the language model head and embedding layers, are optimized. This selective fine-tuning contributes to the model's efficiency by focusing on task-relevant parts of the network. Moreover, the inclusion of an embedding token (the specialized token <embedding>) enables the model to transform the generated detailed image descriptions, aligned with user-defined criteria, into embeddings that are biased toward user interests. These embeddings form the backbone of the relational graph, where nodes represent data instances, and edges encode the similarities between them.

```
import torch
import torch.nn as nn
import networkx as nx
from llava.model import LlavaQwenForCausalLM
from peft import LoraConfig, get_peft_model

class MultipleClusteringLlavaQwen(nn.Module):
    # Model definition
    def __init__(self):
        super(MultipleClusteringLlavaQwen, self).__init__()
        # MLLM initialization
        self.llm = LlavaQwenForCausalLM.from_pretrained(
            "LLaVA-Video-7B-Qwen2",
            attn_implementation="flash_attention_2",
            torch_dtype=torch.bfloat16,
            low_cpu_mem_usage=False,
        )
        self.llm.get_model().initialize_vision_modules()
        vision_tower = self.llm.get_vision_tower()
        vision_tower.to(dtype=torch.bfloat16)

        # LORA configuration
        lora_config = LoraConfig(
            r=64,
            lora_alpha=16,
            target_modules=self.find_all_linear_names(),
            lora_dropout=0.05,
            bias="none",
            task_type="CAUSAL_LM",
        )
        self.llm = get_peft_model(self.llm, lora_config)
        self.llm.base_model.model.lm_head.requires_grad_(True)
        self.llm.base_model.model.model.embed_tokens.requires_grad_(True)
        self.llm.base_model.model.model.mm_projector.requires_grad_(True)

        # Embedding token definition
        self.tokenizer = AutoTokenizer.from_pretrained(
```

```
864              "LLaVA-Video-7B-Qwen2",
865              cache_dir=None,
866              model_max_length=32768,
867              padding_side="right",
868              use_fast=True,
869          )
870          self.tokenizer.add_tokens(["<embedding>"], special_tokens=True)
871          self.embedding_token_index = self.tokenizer.convert_tokens_to_ids
             ("<embedding>")

872
873          # Embedding projection layer
874          self.ln_final = nn.LayerNorm(3584)
875          self.emb_proj = nn.Parameter(torch.empty(3584, 768))
876          nn.init.normal_(self.emb_proj, std=3584 ** -0.5)
877          self.logit_scale = nn.Parameter(torch.ones([]) * np.log(1 / 0.07)
             )

878
879          # Embedding similarity threshold to filter weak connections
880          self.embedding_similarity_threshold = 0.6
881          # Number of reprentative nodes for each cluster
882          self.num_centroids = 3
883          # Number of candidate neighbouring nodes for each cluster
             self.num_candidates = 1

884
885          # MLLM Embedding list
886          self.embedding_list = []
887          # Agent neighbouring node membership assessment dict
888          self.agent_local_assessment_dict = {}
889          # Agent neighbouring cluster merge assessment dict
890          self.agent_global_assessment_dict = {}
891          # Relational graph for agent traversal
892          self.relational_graph = nx.Graph()

893          # Connected components for the relational graph
894          self.communities = []
895          # Current cluster for each connected component
896          self.local_clusters = []
897          # Reprentative nodes for each current cluster
898          self.local_centroids = []
899          # Candidate neighboring nodes for each current cluster
             self.local_candidates = []

900          # Searched clusters from different connected components
901          self.global_clusters = []
902          # Reprentative nodes for each searched cluster
903          self.global_centroids = []
904          # Candidate neighboring clusters for each searched cluster
             self.global_candidates = []

905      # Find LORA layers
906      def find_all_linear_names(self):
907          cls = torch.nn.Linear
908          lora_module_names = set()
909          multimodal_keywords = ["mm_projector", "vision_tower", "
             vision_resampler"]
910          for name, module in self.llm.named_modules():
911              if any(mm_keyword in name for mm_keyword in
                 multimodal_keywords):
912                  continue
913              if isinstance(module, cls):
914                  lora_module_names.add(name)
915          if "lm_head" in lora_module_names:  # needed for 16-bit
916              lora_module_names.remove("lm_head")
917
```

```
        return list(lora_module_names)
```
Listing 2: Multi-Modal Large Language Model Code Snippet

## B.3 MLLM-BASED IMAGE EMBEDDING EXTRACTION

The MLLM-Based Image Embedding Extraction process is crucial for generating user-interest-biased image representations in the agent-centric personalized clustering framework. As detailed in the provided code snippet, the model takes input data, including images and user-defined criteria, and generates detailed image descriptions aligned with those interests. The hidden states corresponding to the <embedding> token in the generated descriptions serve as image embeddings, representing the images in a compact form that reflects user preferences.

The embeddings are further refined through layer normalization and projection to align them with user-defined criteria during training. They are then normalized to facilitate similarity calculation and relational graph construction, supporting efficient agent-centric graph traversal for personalized clustering. In contrast to CLIP-based embeddings, which are constrained by coarse-grained modality alignment, this approach leverages the MLLM's reasoning capabilities to generate contextually rich embeddings that focus on user-specified aspects and attend to relevant visual details, thereby improving clustering accuracy.

```python
import torch
import torch.nn as nn

class MultipleClusteringLlavaQwen(nn.Module):
    # Extract user-interest-biased embeddings
    def extract_embeddings(self, data):
        result = self.llm.generate(
            inputs=data["input_ids"],
            images=data["images"],
            image_sizes=data["image_sizes"],
            modalities=data["modalities"],
            position_ids=None,
            attention_mask=data["attention_mask"],
            do_sample=True,
            temperature=0.2,
            top_p=None,
            num_beams=1,
            # no_repeat_ngram_size=3,
            max_new_tokens=2048,
            use_cache=True,
            return_dict_in_generate=True)

        output_ids = result["sequences"]

        img_embeddings = torch.cat([x[-1] for x in result["hidden_states"
]], dim=1)
        img_embeddings = img_embeddings[:, -output_ids.shape[1]:]
        img_embeddings = self.ln_final(img_embeddings)
        img_embeddings = img_embeddings @ self.emb_proj
        img_embeddings = F.normalize(img_embeddings, p=2, dim=-1)

        logit_scale = self.logit_scale.exp().clamp(max=100).cpu().numpy()

        for i in range(len(output_ids)):
            embedding_index = torch.where(output_ids[i] == self.
embedding_token_index)[0]
            if len(embedding_index) > 0:
                img_embedding = img_embeddings[i][embedding_index[0] + 1]
                img_embedding = img_embedding.detach().float().cpu().
numpy()
            else:
                img_embedding = None
```

```
            self.embedding_list.append({
                "sample_id": data["sample_id"],
                "img_embedding": img_embedding,
                "logit_scale": logit_scale,
            })
```

Listing 3: MLLM-Based Image Embedding Extraction Code Snippet

### B.4 MLLM-BASED IMAGE EMBEDDING TRAINING

The MLLM-Based Image Embedding Training process is designed to fine-tune image embeddings in alignment with user interests, thereby enabling more accurate and efficient clustering. As outlined in the provided code snippet, this process involves training the Multi-Modal Large Language Model (MLLM) on image pairs, where each image is associated with a caption (a detailed description based on user interests) generated by GPT-4, and each pair is assigned a binary similarity pseudo label inferred from GPT-4's similarity assessment.

During training, the model takes each image and user-defined criteria as input, with GPT-4's image caption (appended with an `<embedding>` token) serving as the target for the MLLM to regress. This is supervised using cross-entropy loss. The hidden states corresponding to the `<embedding>` token in the target are extracted as image embeddings, acting as the user-interest-biased representation of the image. These embeddings are refined through layer normalization and projection, then further normalized to ensure they are on the correct scale for similarity calculations. The similarities between the two image embeddings are computed using cosine similarity, scaled by the learnable logit_scale parameter, and supervised with GPT-4's pseudo labels via binary cross-entropy loss. This contrastive-style supervision ensures that the image embeddings are fine-tuned to reflect the user interests, improving their effectiveness in downstream clustering tasks.

```python
import torch
import torch.nn as nn

class MultipleClusteringLlavaQwen(nn.Module):
    # Train user-interest-biased embeddings
    def train_embeddings(self, data):
        outputs1 = self.llm(
            input_ids=data["caption1"]["input_ids"],
            attention_mask=data["caption1"]["attention_mask"],
            labels=data["caption1"]["labels"],
            images=data["caption1"]["images"],
            modalities=data["caption1"]["modalities"],
            image_sizes=data["caption1"]["image_sizes"],
            output_hidden_states=True,
            return_dict=True)

        outputs2 = self.llm(
            input_ids=data["caption2"]["input_ids"],
            attention_mask=data["caption2"]["attention_mask"],
            labels=data["caption2"]["labels"],
            images=data["caption2"]["images"],
            modalities=data["caption2"]["modalities"],
            image_sizes=data["caption2"]["image_sizes"],
            output_hidden_states=True,
            return_dict=True)

        img_embeddings1 = self.ln_final(outputs1["hidden_states"][-1]) @
    self.emb_proj
        img_embeddings2 = self.ln_final(outputs2["hidden_states"][-1]) @
    self.emb_proj

        img_embeddings1 = F.normalize(img_embeddings1, p=2, dim=-1)
        img_embeddings2 = F.normalize(img_embeddings2, p=2, dim=-1)

        img_embeddings1 = img_embeddings1[torch.where(data["caption1"]["
    labels"] == self.embedding_token_index)]
```

```
1026        img_embeddings2 = img_embeddings2[torch.where(data["caption2"]["
1027    labels"] == self.embedding_token_index)]
1028
1029        logit_scale = self.logit_scale.exp().clamp(max=100)
1030        logits = torch.einsum("bd,bd->b", img_embeddings1,
1031    img_embeddings2)
1032        logits = logit_scale * logits
1033
1034        loss_similarity = F.binary_cross_entropy_with_logits(logits, data
        ["img_similarity_label"])
1035
1036        loss_dict = {
1037            "caption": outputs["loss"],
1038            "similarity": loss_similarity,
1039        }
        return loss_dict
```

Listing 4: MLLM-Based Image Embedding Training Code Snippet

### B.5 MLLM-Embedding-Based Graph Construction

To enable efficient agent traversal and clustering, we construct a relational graph in which nodes represent input images and edges encode the pairwise similarity between their corresponding embeddings extracted by the MLLM. As shown in the provided code snippet, we first extract all valid embeddings and compute their cosine similarities, scaled by the logit_scale parameter to adjust the sharpness of similarity scores. A sigmoid function is then applied to map raw similarity values into the range $[0, 1]$. To ensure graph sparsity and remove noisy or semantically weak connections, we filter out edges whose similarity scores fall below a predefined threshold.

A relational graph is then constructed by adding each image (identified by its sample_id) as a node. High-confidence edges—those with similarity scores exceeding the threshold—are added between node pairs, with edge weights corresponding to their computed similarity. This results in a sparse, high-quality relational graph that preserves meaningful semantic relationships while eliminating spurious connections. By retaining only a compact set of high-quality neighbors, the graph significantly reduces the number of traversal steps required by agents to evaluate them, thereby improving clustering efficiency.

```
1058    import torch
1059    import torch.nn as nn
1060    import numpy as np
1061
1062    class MultipleClusteringLlavaQwen(nn.Module):
1063        def build_relational_graph(self):
1064            # Compute the similarity matrix
1065            img_embeddings = [sample["img_embedding"] for sample in self.
        embedding_list]
1066            valid_embeddings = np.array([embedding for embedding in
        img_embeddings if embedding is not None])
1067            logit_scale = self.embedding_list[0]["logit_scale"]
1068            valid_similarity_matrix = valid_embeddings @ valid_embeddings.T
1069            valid_similarity_matrix = valid_similarity_matrix * logit_scale
1070            valid_similarity_matrix = 1 / (1 + np.exp(-
        valid_similarity_matrix))
1071
1072            # Filter weak and invalid edges
1073            mask = np.array([embedding is not None for embedding in
        img_embeddings])
1074            similarity_matrix = np.zeros((len(img_embeddings), len(
        img_embeddings)))
1075            similarity_matrix[mask[:, None] & mask[None, :]] =
        valid_similarity_matrix.ravel()
1076            similarity_matrix = np.triu(similarity_matrix, k=1)
1077            high_similarity_indices = np.argwhere(similarity_matrix >= self.
        embedding_similarity_threshold)
```

```
        # Construct the relational graph
        for sample in self.embedding_list:
            self.relational_graph.add_node(sample["sample_id"])
        for i, j in high_similarity_indices:
            sample_id1 = self.embedding_list[i]["sample_id"]
            sample_id2 = self.embedding_list[j]["sample_id"]
            similarity_score = similarity_matrix[i, j]
            self.relational_graph.add_edge(sample_id1, sample_id2, weight
    =similarity_score)
```

Listing 5: MLLM-Embedding-Based Graph Construction Code Snippet

## B.6 Agent-Based Node Membership & Cluster Merge Assessment

To ensure that clustering decisions reflect fine-grained, user-specific semantics, we incorporate an agent-based assessment mechanism during both node membership evaluation and cluster merging. This process leverages the reasoning capabilities of the MLLM to guide decisions based on visual content and user-defined criteria.

For node membership assessment, each candidate node and its associated cluster centroids are encoded as visual inputs and passed to the MLLM, along with a prompt designed to elicit a binary decision. The model generates a textual response containing a structured <CONCLUSION> tag indicating whether the candidate should be included in the cluster. A response of "yes" results in a positive assignment (similarity = 1.0), while "no" leads to rejection (similarity = 0.0). Unrecognized or missing conclusions (similarity = −1) are ignored during agent graph traversal.

Similarly, cluster merge assessment involves presenting two sets of cluster centroids as inputs. The MLLM evaluates whether the two clusters represent redundant or semantically similar concepts. The decision, extracted from the <CONCLUSION> tag, determines whether the clusters should be merged or remain separate. These assessments are stored in dictionaries that guide subsequent relational graph updates and cluster refinement during the agent traversal process.

This agent-based assessment framework introduces semantic interpretability and user-interest alignment into the clustering pipeline, enabling more accurate and personalized outcomes.

```
import torch
import torch.nn as nn
import numpy as np

class MultipleClusteringLlavaQwen(nn.Module):
    # Assess candidate node's membership in the local cluster
    def assess_node_membership_with_agents(self, data):
        result = self.llm.generate(
            inputs=data["input_ids"],
            # cluster representative images + candidate image
            images=data["images"],
            image_sizes=data["image_sizes"],
            modalities=data["modalities"],
            position_ids=None,
            attention_mask=data["attention_mask"],
            do_sample=True,
            temperature=0.2,
            top_p=None,
            num_beams=1,
            # no_repeat_ngram_size=3,
            max_new_tokens=2048,
            use_cache=True,
            return_dict_in_generate=True)

        output_ids = result["sequences"]
        stop_str = "<|im_end|>"
        outputs = self.tokenizer.batch_decode(output_ids,
    skip_special_tokens=True)
```

```
1134        for i, output in enumerate(outputs):
1135            output = output.strip()
1136            if output.endswith(stop_str):
1137                output = output[:-len(stop_str)]
1138            output = output.strip()
1139
1140            match = re.search(r"<CONCLUSION>(.*?)</CONCLUSION>", output,
        re.DOTALL)
1141            if match:
1142                conclusion = match.group(1).strip()
1143                if "yes" in conclusion.lower():
1144                    img_similarity = 1.0
1145                elif "no" in conclusion.lower():
1146                    img_similarity = 0.0
1147                else:
1148                    img_similarity = -1.0
                    print(f"No answer found in {conclusion}.")
1149            else:
1150                img_similarity = -1.0
                    print(f"No conclusion found in {output}.")
1151
1152            self.agent_local_assessment_dict.update({
1153                # sample_pair: (local_centroids, candidate_node)
1154                data["sample_pair"]: img_similarity,
                })
1155
1156    # Assess whether the two clusters should be merged
1157    def assess_cluster_merge_with_agents(self, data):
1158        result = self.llm.generate(
1159            inputs=data["input_ids"],
1160            # cluster1 + cluster2 representative images
                images=data["images"],
1161            image_sizes=data["image_sizes"],
1162            modalities=data["modalities"],
1163            position_ids=None,
1164            attention_mask=data["attention_mask"],
1165            do_sample=True,
1166            temperature=0.2,
1167            top_p=None,
                num_beams=1,
1168            # no_repeat_ngram_size=3,
1169            max_new_tokens=2048,
1170            use_cache=True,
                return_dict_in_generate=True)
1171
1172        output_ids = result["sequences"]
1173        stop_str = "<|im_end|>"
1174        outputs = self.tokenizer.batch_decode(output_ids,
        skip_special_tokens=True)
1175        for i, output in enumerate(outputs):
1176            output = output.strip()
1177            if output.endswith(stop_str):
1178                output = output[:-len(stop_str)]
1179            output = output.strip()
1180
1181            match = re.search(r"<CONCLUSION>(.*?)</CONCLUSION>", output,
        re.DOTALL)
1182            if match:
1183                conclusion = match.group(1).strip()
1184                if "yes" in conclusion.lower():
1185                    img_similarity = 1.0
1186                elif "no" in conclusion.lower():
1187                else:
                    img_similarity = -1.0
```

```
1188                    print(f"No answer found in {conclusion}.")
1189            else:
1190                img_similarity = -1.0
1191                print(f"No conclusion found in {output}.")
1192
1193            self.agent_global_assessment_dict.update({
1194                # sample_pair: (global_centroids, candidate_centroids)
1195                data["sample_pair"]: img_similarity,
1196            })
```

Listing 6: Agent-Based Node Membership & Cluster Merge Assessment Code Snippet

## B.7 AGENT-CENTRIC GRAPH TRAVERSAL

Once the relational graph is constructed, we employ agents to traverse the graph to discover clusters aligned with user interests. As illustrated in the following code snippet, each connected component of the graph is treated as an independent subgraph. For each component, we assign an agent to it and initialize a cluster by selecting the node with the highest degree (weighted by edge similarity), which serves as the initial centroid (i.e., representative node). The agents then iteratively expand their respective clusters by evaluating the membership of neighboring nodes.

Candidate neighbors are selected from the neighborhood of the current cluster within its component. An MLLM-based agent evaluates each candidate node against the cluster's centroids based on user-defined criteria to determine whether it should be merged into the cluster. Positive candidates are added to the cluster, and the centroids of the cluster are updated by selecting the top-degree nodes. In contrast, negative candidates lead to the removal of their connecting edges to the cluster, reducing noise and potential interference in future traversal.

This iterative process continues until no more valid neighbors remain. If a cluster is completed, it is pushed to the global list, and the agent initializes a new cluster within the remaining subgraph. This method ensures that the traversal is both efficient and semantically meaningful, as decisions are guided by MLLM-based assessments. The agent-centric traversal strategy enables the discovery of fine-grained, user-interest-aligned clusters while dynamically refining the graph structure to suppress noisy or irrelevant connections.

```
1219    import torch
1220    import torch.nn as nn
1221    import networkx as nx
1222    import numpy as np
1223
1224    class MultipleClusteringLlavaQwen(nn.Module):
1225        # Traverse the graph with agents to search for clusters
1226        def traverse_graph_with_agents(self):
1227            # Construct graph and initialize clusters
1228            if len(self.embedding_list) > 0:
1229                # Construct relational graph
1230                self.build_relational_graph()
1231                # Find connected components of the graph
1232                self.communities = list(nx.connected_components(self.
        relational_graph))
1233                self.communities = [community for community in self.
        communities if len(community) > 0]
1234                # Initialize a cluster for each component
1235                self.local_clusters = [[
1236                    max(self.relational_graph.subgraph(community).degree(
        weight="weight"), key=lambda x: x[1])[0]
1237                    ] for community in self.communities]
1238                # Initialize representative nodes for each cluster
1239                self.local_centroids = [tuple(cluster) for cluster in self.
        local_clusters]
1240                # Reset MLLM embedding list
1241                self.embedding_list = []

        # Update local clusters and relational graph
```

```python
        if len(self.agent_local_assessment_dict) > 0:
            for i, (local_cluster, local_centroids, local_candidates) in
enumerate(zip(self.local_clusters, self.local_centroids, self.
local_candidates)):
                if local_candidates is None:
                    continue

                pos_candidates = []
                neg_candidates = []
                for candidate in local_candidates:
                    if self.agent_local_assessment_dict[(local_centroids,
 candidate)]:
                        pos_candidates.append(candidate)
                    else:
                        neg_candidates.append(candidate)

                if len(pos_candidates) > 0:
                    # update local cluster
                    local_cluster.extend(pos_candidates)
                    self.local_clusters[i] = local_cluster
                    # update local centroids
                    num_centroids = min(self.num_centroids, len(
local_cluster))
                    subgraph = self.relational_graph.subgraph(
local_cluster)
                    local_centroids = tuple([node for node, degree in
sorted(subgraph.degree(weight="weight"), key=lambda x: x[1], reverse=
True)[:num_centroids]])
                    self.local_centroids[i] = local_centroids

                if len(neg_candidates) > 0:
                    # update relational graph
                    edges_to_remove = list(nx.edge_boundary(self.
relational_graph, local_cluster, neg_candidates))
                    self.relational_graph.remove_edges_from(
edges_to_remove)

            # Reset node membership assessment dict
            self.agent_local_assessment_dict = {}

        # Find neighbors and select candidate for each local cluster
        self.local_candidates = []
        for i, (local_cluster, local_centroids, community) in enumerate(
zip(self.local_clusters, self.local_centroids, self.communities)):
            # find local neighbors
            local_neighbors = self.find_local_cluster_neighbors(
local_cluster, community)
            while len(local_neighbors) == 0 and len(community) != 0:
                # push local cluster to global clusters
                self.global_clusters.append(local_cluster)
                self.global_centroids.append(local_centroids)
                # remove local cluster from communities
                community = community - set(local_cluster)
                if len(community) > 0:
                    subgraph = self.relational_graph.subgraph(community)
                    local_cluster = [max(subgraph.degree(weight="weight")
, key=lambda x: x[1])[0]]
                    local_centroids = tuple(local_cluster)
                    local_neighbors = self.find_local_cluster_neighbors(
local_cluster, community)
                else:
                    local_cluster = local_neighbors = []
                    local_centroids = tuple(local_cluster)
            self.local_clusters[i] = local_cluster
            self.local_centroids[i] = local_centroids
```

```
            self.communities[i] = community
            self.local_candidates.append(local_neighbors if len(
    local_neighbors) > 0 else None)

        # Generate sample pairs for node membership assessment
        sample_pairs = []
        for i, (local_centroids, local_candidates) in enumerate(zip(self.
    local_centroids, self.local_candidates)):
            if local_candidates is not None:
                for candidate in local_candidates:
                    sample_pairs.append((local_centroids, candidate))
        return sample_pairs

    # Find neighbors of the local cluster
    def find_local_cluster_neighbors(self, local_cluster, community):
        local_cluster = set(local_cluster)
        subgraph = self.relational_graph.subgraph(community)
        neighbors = nx.node_boundary(subgraph, local_cluster)

        if len(neighbors) > self.num_candidates:
            subgraph = self.relational_graph.subgraph(local_cluster |
    neighbors)
            neighbors = [node for node, degree in sorted(subgraph.degree(
    weight="weight"), key=lambda x: x[1], reverse=True) if node in
    neighbors]
            neighbors = neighbors[:self.num_candidates]
        return neighbors
```
Listing 7: Agent-Centric Graph Traversal Code Snippet

## B.8   AGENT-CENTRIC CLUSTER MERGE

After local clusters are formed through agent-centric graph traversal with each connected component, we perform a global refinement step to merge semantically redundant clusters. As shown in the provided code snippet, each cluster is represented by a set of centroids (i.e., representative nodes), and the model identifies neighboring clusters as merge candidates by evaluating inter-cluster connectivity. For each candidate cluster pair, an MLLM-based agent assesses whether the two clusters should be merged based on user-defined criteria. If deemed redundant, the clusters are merged; otherwise, connecting edges are removed to reinforce separation.

To maintain index consistency, cluster pairs selected for merging are processed in reverse order of their indices. Once merged, the new cluster's centroids are updated by selecting the top-degree nodes within the merged subgraph. This merging process reduces redundancy and corrects connectivity errors that may have occurred during local graph traversal. By leveraging the MLLM's reasoning capabilities in the global stage, the framework produces clusters that are both well-separated and structurally coherent, while remaining closely aligned with fine-grained user interests.

```
import torch
import torch.nn as nn
import networkx as nx
import numpy as np

class MultipleClusteringLlavaQwen(nn.Module):
    # Merge semantically redundant clusters with agents
    def merge_clusters_with_agents(self):
        # Update global clusters and relational graph
        if len(self.agent_global_assessment_dict) > 0:
            # Identify indices of cluster pairs to be merged
            to_merge = set()
            for i, (global_cluster, global_centroids, global_candidates)
    in enumerate(zip(self.global_clusters, self.global_centroids, self.
    global_candidates)):
                if global_candidates is None:
                    continue
```

```
1350              candidate_index = self.global_centroids.index(
1351     global_candidates)
1352              if candidate_index == i:
1353                  continue  # Avoid self-merging
1354
1355              if self.agent_global_assessment_dict[(global_centroids,
1356     global_candidates)]:
1357                  to_merge.add(tuple(sorted((i, candidate_index))))
1358              else:
1359                  edges_to_remove = list(nx.edge_boundary(self.
1360     relational_graph, global_cluster, self.global_clusters[
1361     candidate_index]))
1362                  self.relational_graph.remove_edges_from(
1363     edges_to_remove)
1364
1365          # Delete in reverse order to avoid index shifting
1366          removed = set()
1367          for i, candidate_index in sorted(to_merge, key=lambda x: x
1368     [1], reverse=True):
1369              if i in removed or candidate_index in removed:
1370                  continue
1371
1372              merged_cluster = list(set(self.global_clusters[i]) | set(
1373     self.global_clusters[candidate_index]))
1374              self.global_clusters[i] = merged_cluster
1375
1376              num_centroids = min(self.num_centroids, len(
1377     merged_cluster))
1378              subgraph = self.relational_graph.subgraph(merged_cluster)
1379              self.global_centroids[i] = tuple(
1380                  node for node, degree in sorted(subgraph.degree(
1381     weight="weight"), key=lambda x: x[1], reverse=True)[:num_centroids])
1382
1383              # Delete the larger index c
1384              self.global_clusters.pop(candidate_index)
1385              self.global_centroids.pop(candidate_index)
1386              self.global_candidates.pop(candidate_index)
1387              removed.add(candidate_index)  # Record deleted index
1388
1389          # Reset cluster merge assessment dict
1390          self.agent_global_assessment_dict = {}
1391
1392      # Find neighbors and select candidate for each global cluster
1393      self.global_candidates = []
1394      for i, (global_cluster, global_centroids) in enumerate(zip(self.
1395     global_clusters, self.global_centroids)):
1396          # find global neighbors
1397          global_neighbors = self.find_global_cluster_neighbors(
1398     global_cluster, global_centroids)
1399          self.global_candidates.append(global_neighbors if len(
1400     global_neighbors) > 0 else None)
1401
1402      # Generate sample pairs for cluster merge assessment
1403      sample_pairs = []
      for i, (global_centroids, global_candidates) in enumerate(zip(
     self.global_centroids, self.global_candidates)):
          if global_candidates is not None:
              sample_pairs.append((global_centroids, global_candidates)
     )
      return sample_pairs

  # Find candidates of the global cluster to merge
  def find_global_cluster_neighbors(self, global_cluster,
     global_centroids):
      neighbors = []
```

```
        degrees = []
        for centroids, cluster in zip(self.global_centroids, self.
    global_clusters):
            if centroids == global_centroids:
                continue
            degree = sum(self.relational_graph.edges[u, v]["weight"] for
    u, v in nx.edge_boundary(self.relational_graph, global_cluster,
    cluster))
            neighbors.append(centroids)
            degrees.append(degree)
        if len(neighbors) > 0:
            neighbors = [neighbor for neighbor, degree in sorted(zip(
    neighbors, degrees), key=lambda x: x[1], reverse=True)]
            neighbors = neighbors[0]
        return neighbors
```

Listing 8: Agent-Centric Cluster Merge Code Snippet

## C  ADDITIONAL EXAMPLES OF MODEL INPUTS AND OUTPUTS

### C.1  EXAMPLES OF AGENT-BASED NODE MEMBERSHIP ASSESSMENT

Table 4 presents additional qualitative examples illustrating the interpretability of our agent-based node membership assessment across different clustering aspects, including number, suits, and color. Each example shows a set of cluster centroids and a candidate node, along with the corresponding prompt and MLLM-generated output. The agent evaluates the candidate's membership by reasoning over visual cues relevant to the specified aspect. For instance, in the "Number" and "Suits" aspects of playing cards, the agent focuses on rank or suit similarities, respectively, while ignoring irrelevant attributes. In the "Color" aspect of fruit images, the agent attends to subtle hue variations. These examples demonstrate the MLLM's ability to generate interpretable decisions aligned with user-defined criteria, validating the semantic alignment and explainability of our clustering process.

It is worth noting that incorrect edges in the relational graph are corrected through agents' graph traversal, as negative assessments lead to edge removals, progressively refining the graph structure and improving clustering purity.

### C.2  EXAMPLES OF AGENT-BASED CLUSTER MERGE ASSESSMENT

Table 5 presents qualitative examples of agent-based cluster merge assessment across different clustering aspects, including number, suits, and color. Each case includes a set of cluster centroids, candidate centroids, a task-specific prompt, and the corresponding MLLM-generated decision. The agent evaluates whether the two clusters should be merged based on user-defined clustering preference. For instance, in the "Number" and "Suits" aspects, the agent compares rank or suit consistency across clusters, while in the "Color" aspect, it focuses on visual color alignment. The structured output includes a human-interpretable explanation and a final decision marked with a <CONCLUSION> tag. These examples demonstrate the model's ability to make semantically meaningful merge decisions and reinforce the interpretability of our agent-based framework.

It is worth noting that missing edges in the relational graph are corrected through agents' cluster merging, as positive assessments trigger the integration of semantically redundant clusters, enhancing the cohesion and completeness of the clustering structure.

Table 4: Examples of Agent-Based Node Membership Assessment.

| Aspect | Prompt | Cluster Centroids | Candidate | MLLM Generation |
|---|---|---|---|---|
| Number | Determine whether the candidate playing card should be included in the existing playing card cluster based on **rank**. Ignore the suits and focus only on rank comparison. Respond with <CONCLUSION> YES </CONCLUSION> or <CONCLUSION> NO </CONCLUSION>. |  |  | The current cluster consists of cards with the rank 9. The candidate card is ranked 6. Based on rank difference, it should not be included. The answer is <CONCLUSION> **NO** </CONCLUSION>. |
| | |  |  | The cluster contains cards of rank 3. The candidate card is also of rank 3. Based on rank similarity, it should be included. The answer is <CONCLUSION> **YES** </CONCLUSION>. |
| | |  |  | The cluster contains cards of rank 4. The candidate card is also of rank 4. Based on rank similarity, it should be included. The answer is <CONCLUSION> **YES** </CONCLUSION>. |
| Suits | Determine whether the candidate playing card should be included in the existing playing card cluster based on **suits**. Ignore the rank and focus only on suits comparison. Respond with <CONCLUSION> YES </CONCLUSION> or <CONCLUSION> NO </CONCLUSION>. |  |  | The cluster contains cards of the club suit. The candidate card is also a club. Based on suit similarity, it should be included. The answer is <CONCLUSION> **YES** </CONCLUSION>. |
| | |  |  | The cluster contains cards of the diamond suit. The candidate card is also a diamond. Based on suit similarity, it should be included. The answer is <CONCLUSION> **YES** </CONCLUSION>. |
| | |  |  | The cluster contains cards of the heart suit. The candidate card is also a heart. Based on suit similarity, it should be included. The answer is <CONCLUSION> **YES** </CONCLUSION>. |
| Color | Determine whether the candidate fruit should be included in the existing fruit cluster based on **color** similarity. Ignore the species and focus only on color comparison. Respond with <CONCLUSION> YES </CONCLUSION>, or <CONCLUSION> NO </CONCLUSION>. |  |  | The cluster consists of yellow-colored fruits. The candidate fruit also appears yellow. Based on color similarity, it should be included. The answer is <CONCLUSION> **YES** </CONCLUSION>. |
| | |  |  | The cluster consists of green-colored fruits. The candidate fruit is yellow-green, which differs from the cluster's dominant green tone. Based on color difference, it should not be included. The answer is <CONCLUSION> **NO** </CONCLUSION>. |
| | |  |  | The cluster consists of red-colored fruits. The candidate fruit is also red, though darker in tone. Based on color similarity, it should be included. The answer is <CONCLUSION> **YES** </CONCLUSION>. |

Table 5: Examples of Agent-Based Cluster Merge Assessment.

| Aspect | Prompt | Cluster Centroids | Candidate Centroids | MLLM Generation |
|---|---|---|---|---|
| Number | Determine whether the two playing card clusters should be merged based on **rank**. Ignore the suits and focus only on rank comparison. Respond with <CONCLUSION> YES </CONCLUSION> or <CONCLUSION> NO </CONCLUSION>. |  |  | The cluster centroids contain cards of rank 9. The candidate centroids consist of cards of rank 6. Since the ranks differ, the clusters should not be merged. The answer is <CONCLUSION> **NO** </CONCLUSION>. |
| | |  |  | The cluster centroids consist of cards ranked 3. The candidate centroids consist of cards ranked 8. Due to the rank difference, the clusters should not be merged. The answer is <CONCLUSION> **NO** </CONCLUSION>. |
| | |  |  | Both cluster and candidate centroids consist of cards ranked 4. Based on rank similarity, the clusters should be merged. The answer is <CONCLUSION> **YES** </CONCLUSION>. |
| Suits | Determine whether the two playing card clusters should be merged based on **suits**. Ignore the rank and focus only on suits comparison. Respond with <CONCLUSION> YES </CONCLUSION> or <CONCLUSION> NO </CONCLUSION>. |  |  | The cluster centroids consist of cards from the club suit, while the candidate centroids consist of spade suit cards. As the suits differ, the clusters should not be merged. The answer is <CONCLUSION> **NO** </CONCLUSION>. |
| | |  |  | Both the cluster and candidate centroids consist of cards from the diamond suit. As the suits match, the clusters should be merged. The answer is <CONCLUSION> **YES** </CONCLUSION>. |
| | |  |  | The cluster centroids consist of cards from the heart suit, while the candidate centroids consist of cards from the spade suit. Due to suit difference, the clusters should not be merged. The answer is <CONCLUSION> **NO** </CONCLUSION>. |
| Color | Determine whether the two fruit clusters should be merged based on **color** similarity. Ignore the species and focus only on color comparison. Respond with <CONCLUSION> YES </CONCLUSION>, or <CONCLUSION> NO </CONCLUSION>. |  |  | The cluster centroids consist of yellow-colored fruits, while the candidate centroids consist of green-colored fruits. Due to the color difference, the clusters should not be merged. The answer is <CONCLUSION> **NO** </CONCLUSION>. |
| | |  |  | Both the cluster and candidate centroids consist of red-colored fruits. As the colors are consistent, the clusters should be merged. The answer is <CONCLUSION> **YES** </CONCLUSION>. |