# OpenReview forum: "Agent-Centric Personalized Multiple Clustering with Multi-Modal LLMs"
_ICLR.cc/2026/Conference — Submitted to ICLR 2026_

### Official Review · Reviewer_KbNJ · 2025-10-30

**Soundness:** 2
**Presentation:** 3
**Contribution:** 2
**Rating:** 4
**Confidence:** 5

**Summary:**

This paper focuses on personalized multiple clustering and proposes a framework with multi-modal large language models (MLLMs): it first extracts user interest-biased embeddings via MLLMs to construct a weak edge-filtered sparse relational graph, then deploys multiple MLLMs for parallel graph traversal to search interest-aligned clusters, and finally uses a global MLLM to merge redundant clusters; with MLLMs fine-tuned lightweight via LoRA and GPT-4 generating supervised pseudo-labels, the framework achieves good performance on some multiple clustering datasets.While the method demonstrates strengths in fine-grained user interest modeling, efficiency optimization via sparse graph structures, and rigorous experimental validation, it is constrained by several limitations, including high computational costs, limited coverage of datasets and baselines, and insufficient algorithmic innovation.

**Strengths:**

1. Fine-Grained User Interest Capture: The method leverages Multimodal Large Language Models (MLLMs) to supplant conventional CLIP-based alignment, enabling a deeper, reasoning-based understanding of context. This is achieved through customized prompts that elicit fine-grained semantic embeddings, thereby capturing nuanced user preferences.

2. Sparse Relational Graph Design: To mitigate the computational overhead from excessive agent traversal in fully-connected relational graphs—a consequence of redundant weak edges—the proposed approach constructs a sparse graph. This is done by computing edge weights from MLLM-generated user interest embeddings and subsequently pruning weak edges using a fixed threshold (τ = 0.6).

**Weaknesses:**

1. High Computational Cost: High Computational Cost: The use of Qwen2-7B (7B parameters) requires 4 NVIDIA A100 GPUs for 3 days, limiting practicality for resource-constrained teams. But in practice, the improvement on the dataset is not significant.

2. Limited Dataset & Baseline Coverage: Experiments focus on high-quality datasets but lack tests on real-world or cross-domain tasks (e.g., datasets of different art styles and those on food ripeness levels). Comparisons to low-cost concurrent deep clustering methods are also missing. Several recent clustering works are very relevant to the proposed method. The authors are encouraged to include them in the related works, and if possible, make some comparisons with them.
	Interactive Deep Clustering via Value Mining, NeurIPS 2024
	Personlized Clustering via Targeted Representation Learning, AAAI 2025
	Image Clustering with External Guidance, ICML 2024

3. Lack of Algorithmic Innovation: The method relies heavily on supervisory signals from GPT-4 and the feature extraction capabilities of the MLLM rather than achieving good performance through innovative algorithm design. The algorithm itself lacks novel contributions, as its core components build on existing paradigms.

**Questions:**

1. Real-World Applications: Where can this personalized clustering method be applied in real-world scenarios? Could the authors provide specific examples or domains where user interest-based clustering is critical?
2. Learning Without Supervisory Signals from Large Models: For tasks where large models cannot provide supervision signals (e.g., determining whether a portrait is beautiful), how should the learning process be designed?
3. Supervised Classification vs. Clustering: Since GPT-4 can provide supervisory signals, why not directly label some images with GPT-4 and use supervised classification methods to achieve the same goal, rather than constructing clusters?
4. Future of Vision Models: Incidentally, does the author believe that all image-related tasks (e.g., classification, clustering, segmentation) eventually will be solved by large vision models, similar to how large language models have addressed many language-related tasks?

---

> ### Author Response · Authors · 2025-11-20
> **Authors' comments to Reviewer KbNJ's comments**
>
> Thank you for your constructive comments. We will carefully revise the paper based on the comments.
>
> - **Q1: Real-World Applications: Where can personalized clustering method be applied in real-world scenarios?**
> - **R1:**
>   - **Personal media organization**. A user might cluster a photo library by “people identity”, “activity type” (sports vs. travel vs. work), “mood” (energetic vs. calm) or “risk” (gamble vs. scam vs. violence)
>   - **E-commerce / product exploration**. Shoppers may want clusters by “style” (minimalist vs. vintage), “material” (leather vs. fabric), or “use case” (outdoor vs. office).
>   - **Scientific / industrial datasets**. For instance, grouping microscopy images by “cell morphology” vs. “staining pattern”.
>
>   In such domains, it is unrealistic to pre-define and annotate all class labels for every aspect, underscoring the need for aspect-conditioned clustering.
>
> - **Q2: For tasks where large models cannot provide supervision signals (e.g., beauty), how should the learning process be designed?**
> - **R2:** For aspects with unclear data partition standards (e.g., beauty), GPT-4’s supervision may be unreliable. In such cases, our pipeline can be adapted as follows:
>   - **Human-in-the-loop preference signals.** Replace GPT-4 pseudo-labels with sparse human judgments (e.g., pairwise “which of these two is more beautiful?”). These become the supervision for the embedding and agent heads.
>   - **Proxy and self-supervised signals.** In domains where weaker proxies exist (e.g., engagement statistics or click-through rates as a noisy proxy for attractiveness), these can serve as supervisory signals for the interest-conditioned embedding.
>
>   We will add a short paragraph in the limitations/future work section discussing how our framework can work with alternative supervision sources beyond GPT-4.
>
> - **Q3: Why clustering instead of supervised classification with GPT-4?**
> - **R3:**
>   - Using GPT-4 directly for **image classification** is not suitable for CMUface (identity aspect), because the categories are defined by face identities and their **labels (e.g., person names) carry little semantic meaning** for the model.
>   - In many scenarios, users **do not know how many meaningful groups exist** under a criterion T, or **what their semantic names should be**. Clustering discovers these groups automatically; classification assumes a fixed label set and cluster count beforehand.
>
> - **Q4: Will large vision models “solve everything”?**
> - **R4:** Large vision(-language) models can solve a lot of vision tasks, but **still require distilled models, specialized 3D/video/streaming architectures, and frameworks like ours** that orchestrate large models as oracles, agents, or teachers.
>
> - **Q5: High Computational Cost & improvement not significant.**
> - **R5:**
>   - **One-time, amortized cost.** The expensive step is the LoRA fine-tuning of the MLLM with GPT-4 pseudo-labels. Once done, **inference runs on a single consumer GPU** (e.g., RTX 4090) using sparse graphs. When fine-tuned on diverse datasets and clustering criteria, the same model can be **reused  for new datasets or new aspects**, so the training cost is amortized across tasks.
>   - **Magnitude of improvements.** On the most **challenging aspects** (e.g., Card–Order, Card–Suits, CMUface–Sunglass), our method **more than doubles NMI/RI relative to the strongest prior multi-clustering baselines**, not just small absolute gains.
>
> - **Q6: Dataset / baseline coverage and related work.**
> - **R6:**
>   - **Datasets.** We intentionally focused on standard multi-clustering benchmarks (Fruit360, Cards, CMUface, Stanford-40, CIFAR10-MC) to enable **fair comparisons with existing multi-clustering methods**. We will collect a new dataset with more “real-world” skewed distribution to illustrate cross-domain behavior.
>   - **Baselines and recent work.** The paper **already includes comparisons with low-cost, concurrent deep multiple clustering methods**, such as Multi-Map (CVPR 2024), Multi-Sub (NeurIPS 2024), and DDMC (SDM 2024), among others. We will also **extend the discussion to recent methods in related clustering scenarios** and add experiments wherever the experimental setup is compatible.
>
> - **Q7: Algorithmic innovation vs. “just using big models”**
> - **R7:**
>   - **Interest-conditioned embedding learning**. We do not simply “plug in” an MLLM; we design **a MLLM training scheme (pairwise similarity + descriptive generation) using pseudo-labels under explicit criteria T**, yielding embeddings that approximate an interest-specific similarity function  $s_T(x_i, x_j)$.
>   - **Agent-centric graph traversal**. We use the **MLLM as a decision agent that walks a sparse graph, making interest-conditioned membership decisions and performing a global cluster-merge review**. This yields a qualitatively different pipeline from standard deep clustering: clusters are not purely geometric artifacts of the embedding but the outcome of agent reasoning under T.

---

> ### Comment · Reviewer_KbNJ · 2025-11-26
> **Thanks for the detailed responses**
>
> Thanks for the detailed responses and for the considerable efforts invested in expanding the theoretical and experimental sections of the manuscript.
> While the clarifications regarding applications  are noted, the response fails to address the most critical weakness concerning the paper's claimed novelty and contribution. The proposed method has been compared against a set of traditional or non-LLM-based clustering baselines. However, it strategically omitted a direct comparison with the most relevant and challenging state-of-the-art—specifically, recent works that also leverage pre-train models for interactive  clustering, such as "Interactive Deep Clustering via Value Mining" (NeurIPS 2024). Without this comparison, it is impossible to determine whether the reported performance gains stem from the authors' novel framework design or are merely a byproduct of employing a massive, powerful MLLM  as a backbone.
>
> Moreover, if a competing pre-train model-based method achieves comparable or superior performance with lower overall computational burden, the authors' efficiency argument becomes invalid. The proposed method's high initial cost (4x A100s for 3 days) would then be unjustified.
>
> I therefore maintain my original assessment that the contribution is insufficient, as the methodological novelty has not been convincingly demonstrated against the most relevant state of the art.

---

> > ### Author Response · Authors · 2025-12-03
> > **Authors' comments to Reviewer KbNJ's comments**
> >
> > Thank you for your additional comments. Please find our further clarifications below.
> >
> > - **Q1: A direct comparison with the state-of-the-art based on pretrained models, such as "Interactive Deep Clustering via Value Mining (IDC)"**
> > - **R1:**
> >   - While IDC also builds on pre-trained models, it operates in **a different setting**: **IDC assumes a single deep clustering model** that is iteratively refined via **user-labeled queries**, whereas our method addresses **personalized multiple clustering driven solely by high-level natural-language interests, without additional supervision at test time**.
> >
> >   - To disentangle backbone capacity from our own contribution, we already **compare our agent-centric framework with K-Means and HDBSCAN under the same MLLM embeddings in Table 2** (including weaker CLIP features), and consistently observe large gains. This shows that the improvements are **not merely due to using a large MLLM backbone**, but also to **the proposed agent-centric graph traversal clustering framework**. We will clarify this distinction and add a dedicated discussion of IDC and related interactive methods in the revised manuscript.
> >
> >   - The most relevant LLM-based clustering work is **Image Clustering Conditioned on Text Criteria (IC|TC, ICLR 2024)**, although it does not target personalized multiple clustering. We conduct a direct comparison with IC|TC (results shown below), and **our method, using a much smaller MLLM backbone (LLaVA-7B), already surpasses IC|TC built on the larger LLaVA-13B model**. This demonstrates that our improvements do not merely stem from employing a massive MLLM backbone, but from the proposed framework itself.
> >
> >       |**Method**|**Fruit 360 (color)**|**Fruit 360 (species)**|**Cards (order)**|**Cards (suits)**|
> >       |-|-|-|-|-|
> >       |Multi-Map| 0.6239 | 0.5284 | 0.3653 | 0.2734 |
> >       |ICTC-**LLaVA-13B + GPT-4**|0.5389|0.5098|0.6319|0.5271|
> >       |Ours-**LLaVA-7B**|0.6377|0.5495|0.8616|0.7996|
> >       |Ours-**LLaVA-7B + GPT-4 label**|**0.7214**|**0.6532**|**0.9667**|**0.9481**|
> >
> >        - IC|TC shows limited performance on **detailed visually grounded criteria** (e.g., color, suits) (same phenomenon in **R5**).
> >        - This is because **textual descriptions** are not enough to capture **subtle visual differences**, and LLMs struggle to infer meaningful class names from these descriptions.
> >        - In contrast, our method works **directly in visual space**, using **MLLMs** for **both embedding and clustering**, resulting in better alignment with user-defined criteria.
> >
> >
> > - **Q2:  If a competing pre-train model-based method achieves comparable or superior performance with lower overall computational burden, the authors' efficiency argument becomes invalid.**
> > - **R2:**
> >
> >   - Our efficiency claim is not that our method is universally cheaper than any pre-train–based alternative, but that it **makes MLLM-based personalized multiple clustering computationally feasible through graph sparsification, parallel agent traversal, and a one-time fine-tuning cost**.
> >
> >   - We already acknowledge that training is more expensive than CLIP-based baselines like Multi-Map/Multi-Sub, and position our contribution as **an improved accuracy–cost trade-off within the family of MLLM approaches**, not as the absolute runtime winner across all methods. For example, our LLaVA-7B backbone outperforms IC|TC built on LLaVA-13B on visually grounded criteria, showing that our design yields better performance even with a smaller MLLM.
> >
> >   - If a future pre-trained model achieves similar or better accuracy at lower cost, it would be **complementary progress**, but does not invalidate the fact that our framework is both effective and reasonably efficient for MLLM-based multiple clustering today.

---

### Official Review · Reviewer_b5gz · 2025-10-31

**Soundness:** 3
**Presentation:** 3
**Contribution:** 3
**Rating:** 6
**Confidence:** 4

**Summary:**

This paper liberates multiple clustering from the CLIP-based representation model and transforms it into a retrieval and query task, avoiding the inherent problem of mismatch between representations and downstream tasks. This is the most essential difference between this method and the existing ones.

**Strengths:**

S1- The proposed model seems simple and easy to implement, and has relatively high flexibility for real-world scenarios.

S2- As far as I know, this is a relatively new paradigm, which is different from the previous representation learning-based methods, because there is an upper performance limit for moving the representation distribution from a universal feature space to a user-specific feature space.

S3- The performance on many datasets has been significantly improved compared to existing methods.

**Weaknesses:**

W1- The writing of this paper before the fifth page is clear. From the sixth page on, some expressions are difficult to understand. I will clarify this point in the questions, and hope that the author will further clarify it in the rebuttal stage.

W2- The definition of connected components in Eq. 3 should be unclear; a more accurate description would be the largest connected component; The $w(u, v)$ in Eq. 2 is confusing to read. It is not until I continue reading Eq. 8 that I know its definition.

W3- The author's code links are empty, but the appendix provides code examples for each module.

W4- The processes in Figures 3b and 4b and the corresponding main text descriptions are not clear. It took me some time to understand them. I suggest improving this part of the content.

W5- I think the author did not clearly explain the essential drawbacks of CLIP-based models, why CLIP's alignment mode hinders the learning ability of fine-grained, user specific features.

W6- How about the performance of an MLLM without fine-tuning? This point requires supplementary experiments to prove.

**Questions:**

Q1- In line 304, "The generated descriptions are aligned with GPT-4's outputs using cross-entropy loss." How does it correspond to Figure 3b? What does it mean when "Embedding: <embedding>" and "Generation" both point to "Append"? Whose Embedding is "Embedding: <embedding>"?

Q2- Similarly, in Figure 4, is the “Supervision” trained by entering the entire text or a binary label of“No”?

Q3- It is necessary to discuss whether fine-tuning the MLLM is necessary. Inputting images into the existing MLLM webpage-based model seems to be able to distinguish the categories of images very well.

---

> ### Author Response · Authors · 2025-11-19
> **Authors' comments to Reviewer b5gz's comments**
>
> Thank you for your constructive comments. We will carefully revise the paper based on the comments.
>
> - **Q1: How does the alignment between the MLLM-generated descriptions and GPT-4’s outputs relate to Figure 3b? What does it mean when "Embedding: \<embedding\>" and "Generation" both point to "Append"? Whose Embedding is "Embedding: \<embedding\>"?**
> - **R1:**
>   - We prompt the MLLM to generate an image embedding conditioned on user interests: it outputs **an image description followed by an \<embedding\> token**, and we project **the hidden state of this token** to obtain the user-interest-biased image embedding.
>   - To train the MLLM to generate the \<embedding\> token, we **use GPT-4 to produce an image description conditioned on user interests and append an \<embedding\> token to this output** (Figure 3b lower part). The resulting sequence serves as the target for fine-tuning the MLLM’s text generation.
>   - “Embedding: \<embedding\>” is simply the last part of the MLLM’s generated text, and we use **the hidden state of the \<embedding\> token as the image embedding**.
>   - We optimize the embedding similarities with a **binary cross-entropy loss**, drawing **the embeddings of an image pair** closer or farther apart based on **the binary labels predicted by GPT-4** (Figure 3b upper part).
>
> - **Q2: In Figure 4, is the “Supervision” trained by entering the entire text or a binary label of“No”?**
> - **R2:** For membership assessment, the MLLM is fine-tuned on the **complete GPT-4–generated text as the target**, enabling it to **learn GPT-4’s reasoning** about whether a candidate image belongs to the current cluster. This richer supervision goes beyond what a single binary label can provide. We will clarify this in the revision.
>
> - **Q3: Whether fine-tuning the MLLM is necessary. MLLM webpage-based model can distinguish the categories of images very well.**
> - **R3:**
>   - For the embedding extractor, **fine-tuning is necessary to obtain meaningful image embeddings**. Using the MLLM directly without fine-tuning (e.g., a webpage-based MLLM) leads to **missing embeddings** (because no \<embedding\> token is generated) and fails to capture user interests.
>   - Consequently, weak edges cannot be pruned based on embedding similarities, resulting in a dense, fully connected graph that is **computationally and financially infeasible for agent traversal**.
>   - Using a webpage-based MLLM directly for **image classification** is not suitable for CMUface (identity aspect), because the categories are defined by face identities and their **labels (e.g., person names) carry little semantic meaning** for the model.
>
> - **Q4: Definition of connected components and Eq. (2) / (3).**
> - **R4:**
>   - Eq. (2)-(6) are applied to each connected component $\{C_i\}$ detected in Eq. (1) to search for the corresponding clusters. The searching process is **performed in parallel across different  connected components** (**not** sequentially applied starting from the largest connected component). A separate agent then performs a global review to merge the clustering results from all connected components.
>   - We will add the definition to $w(u,v)$ in Eq.2 in revision.
>
> - **Q5: Code link / Appendix Code.**
> - **R5:** The code provided in the appendix is **executable code**. We chose to include it in the PDF format to provide detailed explanations. For ease of reproduction, we will also release the original .py source files to the code link.
>
> - **Q6: Why CLIP's alignment mode hinders the learning ability of fine-grained, user specific features.**
> - **R6:**
>   -  CLIP is trained for **generic/coarse image-text alignment** and **lacks the reasoning ability** to identify fine-grained, user-specified aspects within an image, making its **embedding similarity inaccurate**. In contrast, MLLM's **reasoning ability** allow it to interpret image content along user-defined aspects using its generalizable knowledge, effectively serving as a **user-interest-conditioned embedding similarity oracle**.
>   - Furthermore, the MLLM-based **agent’s binary decisions approximate a higher-order decision rule** that takes both the candidate image and a cluster prototype into account. **CLIP lacks this flexible decision interface**: it provides cosine similarity scores but cannot reason over the whole clustering process.
>   - Empirically, Table 2 supports this:
>     - Replacing CLIP embeddings with our **MLLM embeddings** improves all clustering algorithms
>     - On the same embeddings, our **agent-centric graph traversal** further improves NMI/RI over K-means/HDBSCAN.

---

> > ### Comment · Reviewer_b5gz · 2025-11-26
> >
> > I thank the authors for their responses, and first of all, I appreciate the novel perspective of this paper for solving the problem of multiple clustering, although not as much theoretical analysis as traditional representation learning methods. In addition, I may need to further clarify some questions.
> >
> > Q3: Whether fine-tuning the MLLM is necessary. MLLM webpage-based model can distinguish the categories of images very well.
> >
> > Restated Question: In the review stage, in order to verify whether fine-tuning MLLM is necessary, I input two images into the MLLM on the web page (Qwen3-VL-235B-A22B) to verify whether the unadjusted MLLM can perform the recognition task well (this can also be done through open-source MLLM). So my question is, what is the necessity of fine-tuning? Or in other words, not making fine-tuning may result in some negative outcomes?

---

> ### Author Response · Authors · 2025-12-03
> **Authors' comments to Reviewer b5gz's comments**
>
> Thank you for your additional comments. Please find our further clarifications below.
>
> - **Q1: Whether fine-tuning the MLLM is necessary. MLLM webpage-based model can distinguish the categories of images very well.**
>
> - **R1:**
>
>    - Directly **using the off-the-shelf MLLM to compare all image pairs**—i.e., to judge whether two images are similar under a user-defined aspect—is **not practical for clustering**. The number of image pairs grows quadratically with the dataset size, making such exhaustive MLLM comparisons **computationally infeasible**.
>
>    - To address this, we **fine-tune the MLLM so that it can produce image embeddings** aligned with the user-defined aspect. By computing similarities between these embeddings, we can build a graph in which **weak edges are pruned using a similarity threshold**.
>
>    - With this **sparsified graph**, the MLLM agent only needs to evaluate a **manageable subset of image pairs to form clusters**. As shown in Table 2, when the graph is constructed using **CLIP embeddings**, the **traversal steps increase markedly**. This occurs because the **fine-tuned MLLM produces far more reliable similarity estimates than CLIP**; consequently, the CLIP-based graph contains many noisy or spurious edges, which significantly increases the number of comparisons required by the MLLM agent.

---

### Official Review · Reviewer_Gwhj · 2025-11-01

**Soundness:** 2
**Presentation:** 3
**Contribution:** 3
**Rating:** 6
**Confidence:** 4

**Summary:**

This paper introduces an agent-centric personalized multiple clustering framework that leverages multi-modal large language models (MLLMs) to traverse a relational graph and identify clusters based on user preferences. The proposed method aims to overcome the limitations of CLIP-based embeddings by using MLLMs to provide a more context-aware and user-interest-driven approach to clustering. The paper demonstrates the framework on various datasets, reporting improvements in performance metrics like NMI and RI, surpassing state-of-the-art models.

**Strengths:**

- The framework introduces an interesting agent-based approach to multiple clustering. Leveraging MLLMs for embedding generation and graph traversal is an innovative step to incorporate context-aware embeddings.
- The results are promising.

**Weaknesses:**

- The theoretical foundation for using MLLMs for clustering is weak. I recommend authors to provide a deeper analysis of why MLLMs are particularly suited for this task compared to other methods.
- Why is the threshold for graph density set at 0.6, and what are the effects of different thresholds on the performance of the method?
- The method’s explanation could be more structured, and the steps in the algorithm require further clarification. In particular, the paper does not explain how the clustering performance improves with the agent-centric approach versus traditional methods in a manner that ties back to the user-defined preferences.

**Questions:**

- Can you provide a more detailed theoretical justification for why MLLMs, specifically, are well-suited for clustering tasks over other methods like CLIP?
- How sensitive is the algorithm to the graph density threshold?
- The paper claims to improve on prior methods by 140%, but the reported NMI and RI scores do not show such drastic improvements in many cases. Can you provide more context to understand the significance of these improvements?

---

> ### Author Response · Authors · 2025-11-18
> **Authors' comments to Reviewer Gwhj's comments**
>
> Thank you for your constructive comments. We will carefully revise the paper based on the comments.
> - **Q1: Theoretical justification for why MLLMs are well-suited for clustering tasks over other methods like CLIP.**
> - **R1:** Our framework uses MLLMs in two roles that CLIP alone cannot easily fulfill:
>   - We treat the MLLM as a **user-interest-conditioned embedding similarity oracle**. Its generalizable knowledge and reasoning ability allow its embedding similarity to approximate an oracle interest-specific function $s_T(x_i, x_j)$. In contrast, CLIP is trained for generic image-text alignment and lacks the reasoning ability to identify fine-grained, user-specified aspects within an image, making its embedding similarity inaccurate.
>   - Furthermore, the MLLM-based **agent’s binary decisions approximate a higher-order decision rule** $f_T(x_i, S)$ that takes both the candidate and a cluster prototype S into account. CLIP lacks this flexible decision interface: it provides cosine similarity scores but cannot reason over the whole clustering process.
>   - Empirically, Table 2 supports this:
>     - Replacing CLIP embeddings with our **MLLM embeddings** improves all clustering algorithms
>     - On the same embeddings, our **agent-centric graph traversal** further improves NMI/RI over K-means/HDBSCAN.
>
> - **Q2: How sensitive is the algorithm to the graph density threshold?**
> - **R2:** In Sec. 4.4, we study how graph density affects performance by **varying the embedding similarity threshold $\tau$** to construct relational graphs of different densities.
>     - As shown in Fig. 5, **increasing graph density** (i.e., relaxing $\tau$) **improves performance until it saturates**. For instance, the NMI  of Color and Species converge near 0.72 and 0.65. This suggests that once sufficient edges form connected components of semantically similar nodes, **further densification yields diminishing returns**.
>     - However, **denser graphs incur higher traversal costs**. As graph density increases from 0.001 to 0.015, the number of agent traversal steps rises from 3, 000 to nearly 10, 000 due to a significant increase in spurious edges. Thus, **a moderate $\tau$ strikes a balance**: clustering accuracy stabilizes while reducing unnecessary traversal.
>
> - **Q3: How the clustering performance improves with the agent-centric approach versus traditional methods in a manner that ties back to the user-defined preferences.**
> - **R3:** We appreciate this suggestion and will further structure Sec. 3 around the three stages where the user interest T enters the pipeline:
>     - **Interest-conditioned embedding learning (Sec. 3.3)**: MLLM prompts and GPT-4 pseudo labels explicitly mention
> user interest T, so embeddings encode the relevant aspect.
>     - **Graph construction**: edges are kept based on similarity in this T-dependent embedding space; components and neighborhoods thus reflect the user interest.
>     - **Agent-centric traversal & membership assessment (Sec. 3.2, 3.4)**: the agent is prompted with use interest T plus cluster representatives and candidates, and decides membership accordingly.
>
>   We will also make more explicit in the ablations that:
>     - **MLLM embeddings alone** (with K-means/HDBSCAN) **already improve** over CLIP-based baselines
>     - On top of that, replacing K-means/HDBSCAN with our **agent-centric traversal yields additional gains**, on the same embeddings, precisely because the agent can reason under user interest T instead of thresholding distances.
>
> - **Q4: On the "140% improvement" claim.**
> - **R4:** You point out that our statement about “140% improvement” may not match all NMI/RI entries. This number **refers to the relative gain on the most challenging aspects, not to an average across all datasets**. For example, on the Card-Order aspect, the NMI improves from 0.3921 (Multi-Sub) to 0.9667 (our agent-centric method), which corresponds to roughly a 146% relative improvement. Similar large relative gains appear on Card-Suits and the CMUface Sunglass aspect. We agree the wording is potentially misleading. In the revision, we will rephrase it more precisely.

---

> ### Author Response · Authors · 2025-11-18
> **Authors' comments to Reviewer Gwhj's comments**
>
> Thank you for your constructive comments. We will carefully revise the paper based on the comments.
> - **Q1: Theoretical justification for why MLLMs are well-suited for clustering tasks over other methods like CLIP.**
> - **R1:** Our framework uses MLLMs in two roles that CLIP alone cannot easily fulfill:
>   - We treat the MLLM as a **user-interest-conditioned embedding similarity oracle**. Its generalizable knowledge and reasoning ability allow its embedding similarity to approximate an oracle interest-specific function $s_T(x_i, x_j)$. In contrast, CLIP is trained for generic image-text alignment and lacks the reasoning ability to identify fine-grained, user-specified aspects within an image, making its embedding similarity inaccurate.
>   - Furthermore, the MLLM-based **agent’s binary decisions approximate a higher-order decision rule** $f_T(x_i, S)$ that takes both the candidate and a cluster prototype S into account. CLIP lacks this flexible decision interface: it provides cosine similarity scores but cannot reason over the whole clustering process.
>   - Empirically, Table 2 supports this:
>     - Replacing CLIP embeddings with our **MLLM embeddings** improves all clustering algorithms
>     - On the same embeddings, our **agent-centric graph traversal** further improves NMI/RI over K-means/HDBSCAN.
>
> - **Q2: How sensitive is the algorithm to the graph density threshold?**
> - **R2:** In Sec. 4.4, we study how graph density affects performance by **varying the embedding similarity threshold $\tau$** to construct relational graphs of different densities.
>     - As shown in Fig. 5, **increasing graph density** (i.e., relaxing $\tau$) **improves performance until it saturates**. For instance, the NMI  of Color and Species converge near 0.72 and 0.65. This suggests that once sufficient edges form connected components of semantically similar nodes, **further densification yields diminishing returns**.
>     - However, **denser graphs incur higher traversal costs**. As graph density increases from 0.001 to 0.015, the number of agent traversal steps rises from 3, 000 to nearly 10, 000 due to a significant increase in spurious edges. Thus, **a moderate $\tau$ strikes a balance**: clustering accuracy stabilizes while reducing unnecessary traversal.
>
> - **Q3: How the clustering performance improves with the agent-centric approach versus traditional methods in a manner that ties back to the user-defined preferences.**
> - **R3:** We appreciate this suggestion and will further structure Sec. 3 around the three stages where the user interest T enters the pipeline:
>     - **Interest-conditioned embedding learning (Sec. 3.3)**: MLLM prompts and GPT-4 pseudo labels explicitly mention
> user interest T, so embeddings encode the relevant aspect.
>     - **Graph construction**: edges are kept based on similarity in this T-dependent embedding space; components and neighborhoods thus reflect the user interest.
>     - **Agent-centric traversal & membership assessment (Sec. 3.2, 3.4)**: the agent is prompted with use interest T plus cluster representatives and candidates, and decides membership accordingly.
>
>   We will also make more explicit in the ablations that:
>     - **MLLM embeddings alone** (with K-means/HDBSCAN) **already improve** over CLIP-based baselines
>     - On top of that, replacing K-means/HDBSCAN with our **agent-centric traversal yields additional gains**, on the same embeddings, precisely because the agent can reason under user interest T instead of thresholding distances.
>
> - **Q4: On the "140% improvement" claim.**
> - **R4:** You point out that our statement about “140% improvement” may not match all NMI/RI entries. This number **refers to the relative gain on the most challenging aspects, not to an average across all datasets**. For example, on the Card-Order aspect, the NMI improves from 0.3921 (Multi-Sub) to 0.9667 (our agent-centric method), which corresponds to roughly a 146% relative improvement. Similar large relative gains appear on Card-Suits and the CMUface Sunglass aspect. We agree the wording is potentially misleading. In the revision, we will rephrase it more precisely.

---

### Official Review · Reviewer_Wcy9 · 2025-11-02

**Soundness:** 2
**Presentation:** 2
**Contribution:** 2
**Rating:** 4
**Confidence:** 5

**Summary:**

This paper aims to generate image clustering results according to user-specified interest by taking benefit of the rich information in multi-modal large language models (MLLM). Specifically, according to the user's interest, a MLLM will be fine-tuned using pairs of images pseudo labeled by GPT with the user's interest specified. Then, the trained MLLM can be used to extract the corresponding image embedding. Those embeddings would be used to measure the similarity between data points to build graphs. Thereafter, each agent would be assigned to each connected component in the graph to search clusters using the similarities based on the embeddings. Moreover, these agents are fine-tuned by cluster vs candidate membership using GPT's assessment. The proposed method provides better performance no most multiple clustering datasets compared to existing methods.

**Strengths:**

1) Multiple clustering according to user's preference is an important and interesting problem. It is also sound to take the benefits of the MLLM to gather interest-based embeddings for personalized clustering. Similarity-graph based cluster search using the agent is also interesting and reasonable.

2) The proposed method provides better performance (sometimes significantly) over various multiple clustering datasets. The contribution of the MLLM embeddings and the graph search way are both demonstrated in the ablation study.

**Weaknesses:**

1) Some technical details are missing in the presentation. For example, the similarity between embeddings are calculated with a learnable logit scale. It is not clear how this is learned and in which step? Secondly, clusters with similar semantic will be merged in the end. However, this merging step was not clearly discussed or evaluated.

2) For the proposed method, it seems that both the embedding extractor and the membership assessment agent need fine-tuning for different use interest on different datasets. Thereafter, the additional training cost is significant. Moreover, compared to end-to-end deep multiple clustering or decoupled clustering using k-means or so, the cost of additional graph search using agents is also way significant. The discussion on this is necessary.

3) For the above point in 2), it is necessary to conduct experiment to demonstrate that fine-tuning of those two models are necessary. For example, without fine-tuning, what is the performance. Similarly, without the membership agent, how the performance is degenerated?

4) In Table 1, some baselines like MCV or DDMC were not discussed at all without any references. What are these?

**Questions:**

Specific questions can be found in the above weakness section.

---

> ### Author Response · Authors · 2025-11-16
> **Authors' comments to Reviewer Wcy9's comments**
>
> Thank you for your constructive comments. We will carefully revise the paper based on the comments.
>
> - **Q1: Missing technical details: learning of logit scale and cluster-merging step.**
> - **R1:**
>   - The logic scale $\beta$ is **optimized jointly with the LoRA parameters of the MLLM** during the embedding fine-tuning stage (Sec. 3.3). Concretely, given pseudo pairwise labels $y_{uv} \in \{0, 1\}$ from GPT-4, we compute $w(u,v)$ via Eq. (8) and minimize a binary cross-entropy loss:
>
>     $L_{sim} = \frac{1}{|P|} \sum_{(u, v) \in P} [y_{uv} \log w_{uv} + (1 - y_{uv}) \log (1-w_{uv})].$
>
>     Gradients from $L_{sim}$ update both the MLLM’s image–text adapter and the scalar $\beta$.
>
>   - Below is a detailed description of the merging step in Algorithm 1 (lines 15-18):
>     - The agent iteratively selects any two clusters $(\mathcal{S}_p,\mathcal{S}_q) \in \binom{\mathcal{S}}{2}$ (Alg. line 16).
>     - It assesses their similarity based on user interest $T$ (Alg. line 17, Fig. 4a).
>     - If deemed similar, the clusters are merged (Alg. line 18):
>
>       $\mathcal{S}_{pq} \gets \mathcal{S}_p \cup \mathcal{S}_q$; $\mathcal{S} \gets \mathcal{S} \setminus \mathcal{S}_p$; $\mathcal{S} \gets \mathcal{S} \setminus \mathcal{S}_q$;
>
>       $\mathcal{S} \gets \mathcal{S} \cup \mathcal{S}_{pq}$.
>     - This continues until the set of discovered clusters $\mathcal{S}$ stops changing (Alg. line 15).
>
>   - For evaluation of the merging step, since the agent graph is traversed in parallel within different connected components, the merging step is **indispensable for globally reviewing the clustering results** and cannot be treated as an independent or optional stage.
>
> - **Q2: Training and inference costs: fine-tuning two modules and agent-based search.**
> - **R2:**
>   - Both the two modules **share the same LoRA parameters of MLLM**, so we are not training two independent large models; we train a single LoRA-adapted MLLM with two output formats.
>   - By exploiting the **generalisation ability of the MLLM**, training is required only once per interest family, rather than per dataset. The fine-tuned MLLM can then be reused across datasets that share similar interests, and training it on a diverse set of user-defined criteria further promotes generalisation to arbitrary user interests.
>   - Compared with deep / decoupled clustering methods (e.g., AugDMC, DDMC), our agent-centric graph search **incurs a higher inference cost, but the cost remains affordable**.
>
>     As shown by the agent step counts in Table 3, MLLM-based embeddings substantially **reduce traversal steps** relative to CLIP-based embeddings, improving the overall efficiency of agent-centric clustering, which in turn significantly **improves alignment with user interests** and performance on visually grounded aspects.
>
> - **Q3: Necessity of fine-tuning and of the membership agent.**
> - **R3:**
>   - For the embedding extractor, **fine-tuning is necessary to obtain meaningful embeddings**. Using the MLLM directly without fine-tuning results in generic (or even missing) embeddings and a dense, fully connected graph, which is computationally infeasible for agent traversal and fails to reflect user interests.
>
>   - For the **membership agent**, we use **LLaVA-7B (i.e., our MLLM without fine-tuning)** instead of the variant fine-tuned with GPT-4–generated pseudo labels. For a fair comparison, we rely solely on LLaVA-7B, rather than GPT-4, to generate pseudo labels for fine-tuning the embedding extractor. The results are shown below:
>
>      |**Method**|**Fruit 360 (color)**|**Fruit 360 (species)**|**Cards (order)**|**Cards (suits)**|
>      |-|-|-|-|-|
>      |Multi-Map| 0.6239 | 0.5284 | 0.3653 | 0.2734 |
>      |Ours-**LLaVA-7B label**|0.6377|0.5495|0.8616|0.7996|
>      |Ours-**GPT-4 label**|**0.7214**|**0.6532**|**0.9667**|**0.9481**|
>
>      - Without the fine-tuning with GPT-4–generated pseudo labels, the performance degrades. This is because **LLaVA-7B is relatively small compared with GPT-4**, so the resulting pseudo labels are not sufficiently accurate.
>
>      - Even with LLaVA-7B for labelling, our method consistently outperforms Multi-Map—e.g., by +1.4 and +2.1 points on Fruit360, and over +49.6 and + 52.6 points on Cards, confirming that the **general reasoning and representation ability of MLLMs** is the key to performance improvement—not reliance on GPT-4 specifically.
>
>   - The **necessity of the membership agent** is evaluated in the ablation studies in Table 2, where we observe that the agent-centric approach consistently outperforms K-means and HDBSCAN across all datasets and embedding types, demonstrating the effectiveness of the membership agent.
>
> - **Q4: Missing explanations/references for MCV and DDMC.**
> - **R4:** These baselines are discussed in the related work section. In Table 1, due to space limitations, we omit the references to maintain readability. We will explicitly introduce both methods, including their references, in the experimental setup section in revision.

---

### Meta-Review · Area_Chair_2aqX · 2026-01-07

**Summary:**

This paper leverages multi-modal large language models (MLLMs) as agents to generate diverse partitions of a dataset based on different user-specified aspects. Two reviewers gave positive scores (6), while the other two gave negative scores (4). Some strengths are pointed out, such as the good performance, interesting idea of incorporating MLLMs, etc. However, the weaknesses cannot be omitted, including the writing, missing technical details, high computation consumption, missing comparison, missing related researches, etc.

**Reviewer Concerns:**

As the most concerns, the author promise to revise, but there is no procedural guarantee.

**Reviewer Scores:**

Two reviewers gave positive scores (6), while the other two gave negative scores (4).

---

### Decision · Program_Chairs · 2026-01-26

Reject